

# Probabilistic Seismic Hazard Assessment of Sweden

Niranjan Joshi[1,2], Björn Lund[1,2], and Roland Roberts[1,2]

[1]Department of Earth Sciences, Uppsala University, Sweden
[2]Centre of Natural Hazards and Disaster Science, Uppsala University, Sweden

**Correspondence:** Niranjan Joshi (niranjan.joshi@geo.uu.se)

**Abstract.** Assessing seismic hazard in stable continental regions (SCR) such as Sweden poses unique challenges compared to active seismic regions. With diffuse seismicity, low seismicity rate, few large magnitude earthquakes and little strong motion data, estimating recurrence parameters and determining appropriate attenuation relationships is challenging. This study presents a probabilistic seismic hazard assessment of Sweden based on a recent earthquake catalogue which includes a large number of events with magnitudes ranging from 5.9 to -1.4, enabling recurrence parameters to be calculated for more source areas than in previous studies, and with less uncertainty. Recent ground motion models developed specifically for stable continental regions, including Fennoscandia, are used in logic trees accounting for their uncertainty and the hazard is calculated using the OpenQuake engine. The results are presented in the form of mean peak ground acceleration (PGA) maps at 475 and 2500 year return periods and hazard curves for four seismically active areas in Sweden. We find the highest hazard in the northernmost part of the country, in the post-glacial fault province. This is in contrast to previous studies, which have not considered the high seismic activity on the post-glacial faults. We also find relatively high hazard along the northeast coast and in southwestern Sweden, whereas the southeast and the mountain region to the northwest have low hazard. For a 475 year return period we estimate the highest PGAs to be 0.04-0.05g, in the far north, and for a 2500 year return period it is 0.1-0.15g in the same area. Significant uncertainties remain to be addressed in regards to the SCR seismicity in Sweden, including the homogenization of small local magnitudes with large moment magnitudes, the occurrence of large events in areas with little prior seismicity and the uncertainties surrounding the potential for large earthquakes on the post-glacial faults in northern Fennoscandia.

## 1   Introduction

Earthquakes are known to cause some of the most fatal natural disasters. As short term prediction of where and when large earthquakes occur is currently not routinely possible, and as the disaster development in the event is very rapid, earthquake risk reduction strategies must focus on the mitigation and preparation phases of the disaster cycle (Elliott, 2020). One key component of this is producing an accurate seismic hazard assessment in terms of probable ground motions, which can help identify areas of high hazard and allow for designing appropriate risk reduction strategies.

At plate boundaries, earthquake activity is high and large events can often be associated with identified fault zones. Conversely, stable continental regions, such as Sweden, tend to show a more diffuse geographically distributed pattern of seismicity (Schulte and Mooney, 2005; England and Jackson, 2011). Occurrence rates are low (e.g. Lund et al., 2021) and areas of high hazard can be difficult to define (e.g. Calais et al., 2016). However, as England and Jackson (2011) show, the risk posed





by earthquakes in such intraplate areas is far from negligible. They showed that of the 130 earthquakes with a thousand or more casualties occurring over a 120 year period prior to 2012, about 100 took place in continental interiors. These intraplate earthquakes also caused approximately 75% more total deaths than the plate boundary events.

Historical and instrumental data until 2005 suggest that Sweden can expect on the order of one magnitude 5 earthquake every century and a magnitude 6 earthquake every millennium (Bödvarsson et al., 2006). As the rate of occurrence of these higher magnitude earthquakes is low, the national authority in charge of building codes (Swedish National Board of Housing, Building and Planning - Boverket) has no current plans to implement Eurocode 8 in Sweden (Boverket, personal communication 2020) and there is no official national seismic hazard map. The general increase in risk awareness in society, particularly after large

earthquakes (Tan and Maharjan, 2018), has, however, led to an increased number of requests for seismic hazard information from building companies and corporate site planning projects. For sensitive infrastructure sites such as nuclear power plants, nuclear waste repositories, dams and mines, seismic hazard estimates are required and there is therefore a need to better define the hazard spatially, estimate potential ground motions and investigate associated uncertainties.

In this paper, we review earlier estimates of seismic hazard in Sweden, with a special emphasis on the recent release of the

European Seismic Hazard Model 2020 (Danciu et al., 2021). We then discuss the development of a new probabilistic seismic hazard model for Sweden, which will take into account the large amount of smaller earthquakes recorded by the Fennoscandian seismic networks in the last two decades (e.g. Lund et al., 2021; Veikkolainen et al., 2021; Ottemöller et al., 2018) as well as recent developments in ground motion models (Fülöp et al., 2020; Weatherill and Cotton, 2020). The hazard is calculated using the OpenQuake engine (Pagani et al., 2014) and we produce hazard maps, with mean estimates for peak ground acceleration

(PGA) corresponding to return periods of 475 and 2500 years, and hazard curves for Sweden's most seismically active areas. Finally, we discuss the uncertainties inherent in the models and the potential need for additional approaches to seismic hazard assessments in Sweden.

## 2   Earthquake activity in Sweden

Sweden is located in a stable continental region (SCR) with very few damaging earthquakes (Bödvarsson et al., 2006; Lund

et al., 2021). Most of the country is part of the Fennoscandian shield of the East European Craton, see Figure 1, with Archean and Early Paleoproterozoic rocks in the north that are to a large extent reworked during the Paleoproterozoic Svecokarelian/Svecofennian orogeny (Stephens and Bergman-Weihed, 2020). The eastern part of the country is dominated by Svecokarelian/Svecofennian rocks whereas the southwest contains Meso- and Neoproterozoic rock of the Sveconorwegian orogeny (Stephens and Bergman-Weihed, 2020). The Gulf of Bothnia in the east, and the Baltic Sea further southeast, comprise Mesoproterozoic to

Ediacaran and Lower Paleozoic sedimentary platformal cover rocks of varying thickness. Onshore Finland comprises Archean, Svecokarelian/Svecofennian and Mesoproterozoic rocks whereas further south, the Baltic countries are covered by sedimentary rocks deposited on the East European Craton (Bogdanova et al., 2008). In western Sweden, older basement and Paleozoic cover rocks are affected by the Caledonian orogeny, which created the present day Swedish mountain chain (Stephens and Bergman-Weihed, 2020). Further west, Norway is composed mostly of rocks from the Caledonian and Sveconorwegian orogens onshore



and passive margin deposits offshore (e.g. Corfu et al., 2014; Bingen et al., 2021). To the south, the craton is separated from Phanerozoic Europe by deformation zones through southernmost Sweden (the Sorgenfrei-Tornquist zone) and Denmark (the Trans European Suture Zone), over which major changes in lithospheric, crustal and sediment cover thicknesses occur (e.g. Sandersen et al., 2021). The opening of the Atlantic some 60 Ma ago is the latest large scale tectonic event to significantly affect Fennoscandia (Stephens and Bergman-Weihed, 2020).

Fennoscandia has been subject to repeated glaciations during the Quaternary. The latest, the Weichselian ice sheet, reached its maximum approximately 20,000 years before present (BP) (Svendsen et al., 2004), before retreating and disappearing around 10 ka BP. Large fault scarps in northern Fennoscandia, see Figure 1, have been dated to the final deglaciation phase (Lagerbäck and Sundh, 2008; Smith et al., 2021) and are inferred to have resulted from earthquakes with magnitudes up to $M_W$ 8 (e.g. Lindblom et al., 2015). Glacial isostatic adjustment is still ongoing in Fennoscandia and GNSS measurements show that all of Sweden is rebounding, with a maximum uplift velocity of 10.3 mm/year on the northeast coast, decreasing to about 1 mm/year in southernmost Sweden (Vestøl et al., 2019).

The seismicity in Fennoscandia has been the subject of many studies, Gregersen et al. (2021) provide a recent review of the whole region whereas Lund et al. (2021) focus on recent observations of earthquakes in Sweden, which we briefly review here. The expansion of the Swedish National Seismic Network (SNSN) from 6 to 65 stations during the first decade of the 21st century has provided a wealth of new earthquake observations, increasing the total number of recorded events from about 1,600 prior to the year 2000 to about 12,500 in 2021. Most of the events are small, with the magnitude of completeness estimated at $M_L$ 0.5 within the network. Areas of high seismic activity include the southwestern part of Sweden across Lake Vänern, along the northeast coast and in the far north, see Figure 1. The southwest and northeast coast seismicity agrees well with what was known already from the pre-instrumental period (e.g. Kjellén, 1910), the new data have, however, revealed that activity locally cluster along specific zones and is less diffuse than previously thought. The new observations have significantly changed how seismic activity in the north is viewed. We now know that seismicity is strongly correlated with the mapped post-glacial fault (PGF) scarps, indicating that the faults are still very active although event magnitudes are generally low (Lindblom et al., 2015). One reason for the lack of early observations along the PGFs is the low population density in the inland. On the Pärvie fault (Figure 1) for example, there is only one event in the catalogue prior to 1967, a $M_L$ 2.6 event in 1927. The largest recorded Pärvie event had $M_L$ 3.7 in 1967 (Ahjos and Uski, 1992). Conversely, the Burträsk PGF, on the northeast coast, has earthquake reports going back to the 18th century, with the largest events having magnitudes around $M_L$ 4-4.5 (Ahjos and Uski, 1992) and the recent data shows that it is the most seismically active region in Sweden.

The largest historically recorded earthquake on land in Fennoscandia was the 1819 Lurøy $M_W$ 5.9 event in Nordland, Norway, located about 80 km from the Swedish border (Mäntyniemi et al., 2020). In 1759, a $M_S$ 5.6 event occurred in the waters between Sweden and Denmark (Wood, 1989), and in 1904 a $M_W$ 5.4 event took place just off the Swedish west coast close to the Norwegian border (Bungum et al., 2016). Additional $M_W$ 5+ events have occurred in the regions around Sweden, such as the 2004 Kaliningrad earthquake doublet (Gregersen et al., 2007), the 1892 $M_W$ 5.2 western and the 1834 $M_W$ 5.0 southern Norway events (Ahjos and Uski, 1992). Recently, Olesen et al. (2021) trenched the post-glacial Stuoragurra fault in



northern Norway and found evidence of an event with magnitude about 7 occurring as recently as 700 years ago. This indicates
that the PGFs may still be capable of generating very large earthquakes.

The driving forces of Fennoscandian seismicity have been a debated subject, see Gregersen et al. (2021) for a recent review.
As glacial rebound dominates the deformation rates of the Fennoscandian crust, both vertically and horizontally, GNSS mea-
surements cannot resolve any tectonic-driven deformation (Vestøl et al., 2019). However, evidence such as focal mechanisms
and in-situ stress measurements points to ridge-push from the opening of the Atlantic as the main driving force for seismicity,
potentially moderated by glacially induced stress, sediment load or other regional to local stresses (Gregersen et al., 2021).

## 3  Previous seismic hazard assessments for Sweden

Here we review earlier seismic hazard assessments for Sweden. We will refer to the intercept and slope of the earthquake
frequency-magnitude distribution as the a- and b-values (Gutenberg and Richter, 1944), $M_c$ as the magnitude of completeness
of a frequency-magnitude distribution and $M_{max}$ as the maximum possible magnitude for an area.

The studies by Båth (1979) and Slunga (1979) were the first to estimate seismic hazard in Sweden. They only had access
to some 25 years of instrumental data and thus relied to a large extent on macroseismic data, which is limited in parts of
Sweden due to the low population density. Båth (1979) estimated recurrence rates for earthquakes of varying magnitudes in
a 2°-by-2° grid across Fennoscandia and found the highest hazard on the Norwegian west coast. For Sweden, he estimated a
5% probability in 10 years of an $M_L \geq 5$ event on both the southwest and northeast coasts. Slunga (1979), investigating the
seismic hazard at sites of the then four Swedish nuclear power plants, estimated accelerations of 0.05 – 0.20 g for an annual
probability of $10^{-5}$. The development of the FENCAT catalogue (Ahjos and Uski, 1992), a continuously updated joint regional
catalogue of earthquakes in Northern Europe since 1375, enabled later studies to use a larger, more homogeneous data set for
Fennoscandia.

FENCAT data until 1987 were used for the first Swedish probabilistic seismic hazard assessment (PSHA) work, directed at
site-specific assessments for Sweden's four nuclear power plants (SKI, 1992). Fennoscandian earthquakes south of latitude 61°
were used for seismicity rate information in various combinations of seismic source areas. The lack of strong motion data from
Fennoscandia prompted SKI (1992) to turn to existing Japanese "Standard Response Spectra for rock sites" for ground motion
models (SKI, 1992). The Japanese spectra were modified to fit Swedish earthquake source properties and crustal attenuation
properties. The envelope ground response spectra resulting from the SKI (1992) study are still in use in the Swedish nuclear
industry today, and for a probability of exceedance of $10^{-5}$ per year and a damping of 5%, the estimated PGA is 0.11g for a hard
rock site (SKI, 1992; Larsson and Larsson, 2018). In a review, Lund et al. (2017) found that SKI (1992) may underestimate the
rate of events larger than approximately magnitude $M_W$ 5.2 and recommended an update with modern data and methodologies.

Kijko et al. (1993) used FENCAT and a model that accounts for varying magnitudes of completeness through time in order
to quantify probabilistic seismic hazard in Sweden by estimating the recurrence rates of earthquakes. They found that Sweden
can expect 5.5 earthquakes per year with magnitudes equal to or greater than 2.4, with the maximum expected earthquake
magnitude in southern Sweden being 4.9 for a 615-year time span, and a maximum magnitude of 4.3 in northern Sweden



over 331 years. Mäntyniemi et al. (1993) expanded the Kijko et al. (1993) study to all of Fennoscandia and also included probabilities for non-exceedance of various magnitudes in 1 and 50 years in defined subregions. The results for Sweden were similar to that of Kijko et al. (1993).

The Wahlström and Grünthal (2000) and follow-up Wahlström and Grünthal (2001) studies perform full probabilistic seismic hazard assessments for Fennoscandia using the FENCAT catalogue. Wahlström and Grünthal (2001) apply two different magnitude homogenization schemes. They used two large regions for magnitude completeness estimates, and six different regional and non-regionalized seismic source area models. Combining the recurrence parameters with two different ground motion models in a logic tree, Wahlström and Grünthal (2001) produce a hazard map for Fennoscandia, with 90% probability

of non-exceedance of median horizontal PGA in 50 years, and hazard curves and hazard deaggregations for Sweden, Norway, Denmark and Finland. Their results indicate that the Norwegian west and northwest coasts have the highest hazard in Fennoscandia. The highest hazard in Sweden was found in the southwest, close to the Norwegian border, with an estimated PGA of 0.03g for 475 years return period and 0.08g for 4745 years return period.

Mäntyniemi et al. (2001) produced a seismic hazard map of Fennoscandia by using the site-specific technique of Kijko and

Graham (1998, 1999) and applying it to a grid of Fennoscandian locations, similar to a smoothed seismicity approach (e.g. Danciu et al., 2021). They estimate the recurrence parameters (a- and b- values) and the maximum magnitude, $M_{max}$, from the data in a selected area around a grid point and use those with a ground motion model to calculate the hazard in terms of ground motion probabilities. Mäntyniemi et al. (2001) used a 0.5° by 0.5° grid with a constant b-value of 0.84±0.01 and an $M_{max}$ of 5.8 over the entire region, while the a-value varied with the seismicity included at each grid point. Ground motion models

by Uski and Tuppurainen (1996) and EMSC (1999) were used and the resulting hazard map shows PGA for a 475 year return period, where the highest hazard was obtained at the northwestern coast of Norway with 0.038g (note that western Norway around Bergen was not included in the analysis). The highest PGA in Sweden reached 0.022g both in southwestern Sweden and on the northeast coast.

Bodare and Kulhánek (2006) used the results in Kijko et al. (1993) and Mäntyniemi et al. (1993) together with a relationship

between $M_L$ and the acceleration at the epicentre (Båth, 1980) to infer that seismic hazard at hydropower dams in Northern Sweden is unlikely to exceed 0.04g in the next 50 – 100 years. In two large PSHA projects for the nuclear industry in Finland, analysis of seismicity and definition of seismic source areas was undertaken for Sweden and Finland. The first, the Fennovoima project, assembled seismologists and geologists from Finland and Sweden to perform a full site-specific PSHA for a proposed new nuclear power plant in northern Finland (Korja and Kosonen, 2015; Saari et al., 2015). Using a similar methodology, a

PSHA revision was later performed for the sites of the existing nuclear power plants in Finland (Korja and Kihlman, 2016). The hazard results of these studies have not been made public. On a larger scale, the Global Seismic Hazard Assessment Program (GSHAP, Grunthal and Group, 1999) and the European Seismic Hazard Models 2013 and 2020 (ESHM13 and ESHM20; Woessner et al., 2015; Danciu et al., 2021, respectively) have estimated seismic hazard in Sweden. ESHM20 is described in some detail in Section 3.1 below.





## 3.1 Sweden in the European Seismic Hazard Model 2020

The most recent evaluation of seismic hazard in Sweden comes from the 2020 European Seismic Hazard Model (ESHM20) (Danciu et al., 2021), which assesses earthquake hazard in the Euro-Mediterranean region and builds on the 2013 version of the model (ESHM13; Woessner et al., 2015). ESHM20 applies a fully probabilistic framework homogeneously across the entire pan-European region, while accounting for cross country-border issues. The earthquake data come from the European PreInstrumental earthquake CAtalogue (EPICA, Rovida and Antonucci, 2021), which spans the years 1000 to 1899, and from an extension of the European-Mediterranean Earthquake Catalogue (EMEC, Grünthal et al., 2013) such that it spans the time 1900 to the end of 2014. Magnitudes are harmonized to the moment magnitude $M_W$, the catalogue is declustered and only events with $M_W \geq 3.5$ are included in the magnitude-frequency analysis. For Sweden and the surrounding region within 300 km of Sweden's borders and economic zone, the unified ESHM20 earthquake catalogue contains 212 events, from 1497 to 2014, with magnitudes $3.5 \leq M_W \leq 5.8$.

In order to estimate the recurrence parameters of the magnitude-frequency distribution at the most appropriate spatial distribution, ESHM20 uses a hierarchy of seismic source areas. Fennoscandia is assigned to a single maximum magnitude zone, within which $M_{max}$ is uniform, and two "completeness zones", CSZ, where reporting in each zone is assumed to be spatially homogeneous through time such that the temporal variation in the magnitude of completeness, $M_c$, is the same. At the next levels, Fennoscandia is divided into four "tectonic" source areas, TSZ, and about 20 smaller "area" source zones, ASZ, based on geology, seismicity and an assumption of homogeneous seismicity rate. In these zones, recurrence parameters (a- and b-vales) are calculated using a doubly truncated Gutenberg-Richter distribution, using an automatic maximum likelihood method based on the earthquakes within the zone (Danciu et al., 2017), for zones with more than 30 earthquakes. For ASZs with less than 30 events, the b-value from the TSZ is re-used and the a-value is rescaled from the TSZ with the ratio of the number of complete events in the ASZ and TSZ. If the ASZ does not contain any events above the magnitude of completeness, an a-value is assigned by rescaling the TSZ value with the area ratio of ASZ to TSZ. Uncertainties are estimated using random sampling and discrete approximation of probability distribution methods, giving 16th, 50th and 84th percentile estimates of the median a- and b-values. A tapered Pareto distribution is also defined to describe the recurrence parameters, primarily to provide a faster decaying alternative closer to the maximum magnitude. For Sweden and a 300 km border region, only two ASZs have enough events for separate estimates of the recurrence parameters, all other use the TSZ b-values and rescaled a-values. All seismogenic source models are implemented through a logic-tree, with the median, 5th and 95th percentiles describing the branches of individual area source Gutenberg-Richter parameters, with 0.6, 0.2 and 0.2 as the respective branch weights. $M_{max}$ in Fennoscandia has three branches, where $M_{max}$ is 6.3, 6.6 and 6.9, with weights 0.5, 0.4 and 0.1, respectively.

ESHM20 uses a scaled backbone approach for implementing ground motion models (GMMs). This is done by selecting a relevant GMM from the literature, and modifying it based on the observations and knowledge of the region's crustal properties to account for the uncertainty in the expected ground motions. The scaled backbone GMM for the cratonic region is obtained by generating median ground motions on hard rock for various magnitude ($4.0 \leq M_w \leq 8.0$) and distance ranges ($1.0 \leq R_{rup}(km) \leq 500km$), starting with equally weighing 21 Central and Eastern United States (CEUS) GMMs and fitting





to a parametric GMM of the same form as Kotha et al. (2020), mapped into a scaled backbone logic tree. These ground motions are defined for very hard rock ($V_{S,30}$ = 3,000 m/s) conditions and re-scaled to reference $V_{S,30}$ = 800 m/s rock condition using the CEUS site amplification models of Stewart et al. (2020) and Hashash et al. (2020). They calibrate the model's epistemic uncertainties and propose a logic tree for the application of the model to cratonic regions where one branch is the new parametric Craton model and one branch an adaption of the Shallow Crustal seismicity model used for Central Europe, only retaining the mid and upper (i.e. higher velocity) twigs of that branch. For most of Sweden, Finland and the Baltic, this logic tree combination for cratonic regions is used, while the Shallow Crustal logic tree is used for most of Norway and Denmark.

The final hazard map shows that for Fennoscandia, the seismic hazard is highest along the Norwegian west coast. The Swedish northeast coast and the coast of the Baltic countries also have relatively high hazard whereas most of Denmark, the interior of north and central Sweden and south-central Finland have the lowest hazard. The highest estimated mean PGA for Sweden was 0.025g for a return period of 475 years, and 0.11g for a return period of 2500 years.

## 4 Methods

Probabilistic seismic hazard assessment (PSHA) is a widely used technique to estimate seismic hazard which we utilize in this study. It quantifies the rate at which ground-motions are expected for all possible earthquake scenarios (Cornell, 1968). The method can be broadly divided into three steps : (1) Preparing earthquake data and associating it with seismic source areas and fault zones, with a calculation of recurrence parameters. (2) Selecting appropriate ground motion models that represent the attenuation of seismic waves. (3) Estimating hazard probabilistically by accounting for the epistemic uncertainties through a logic tree. Here we detail how our study implements these steps.

### 4.1 Data preparation

A number of seismic networks in Fennoscandia collect seismic data and report to FENCAT and other international compilations: The Swedish National Seismic Network (SNSN, Lund et al., 2021), The Norwegian National Seismic Network (NNSN, Ottemöller et al., 2018) and NORSAR (Norsar, 1971), The Finnish National Seismic Network (FNSN, Veikkolainen et al., 2021; Ahjos and Uski, 1992) and Oulu University (Sodankylä Geophysical Observatory / University of Oulu , 1980), the Geological Survey of Denmark and Greenland (GEUS, 2023) and the Geological Survey of Estonia (Soosalu et al., 2022). We compile a homogenized high-quality earthquake catalogue of events in Sweden and a surrounding 300 km zone, using FENCAT, the full SNSN catalogue and the ESHM20 unified earthquake catalogue. The data is processed with the methodologies developed in Uski et al. (2015), comprising (i) data merging and cleaning, (ii) declustering, and (iii) magnitude homogenization. We briefly describe these steps below.

### 4.1.1 Data merging and cleaning

The version of the FENCAT catalogue at our disposal spans the year 1375 until the end of 2014 (Korja and Kihlman, 2016). We added the semi-reviewed FENCAT between 2015-01-01 and 2020-09-30 (Finnish National Seismic Network, 2023). Both





these catalogues contain earthquakes from all over Fennoscandia and Northern Europe. The SNSN does not report all small microearthquakes to FENCAT, so we also added the entire 2000-08-15 to 2022-02-28 SNSN catalogue to the data set. In addition, we extracted a few events in the region of interest from the unified ESHM20 catalogue (Danciu et al., 2021). Duplicates were removed from the resulting list of events and remaining non-tectonic events such as quarry, industrial or military blasts, rock bursts, mine collapses etc were removed from the data. As FENCAT contains events far from Sweden, such as along the

Mid-Atlantic ridge, we removed these events together with events in Britain, central Poland, Russia east of 40° longitude and Svalbard prior to further processing. The final earthquake catalogue comprises 24,215 events within the tectonic source areas (Section 4.2) of the hazard assessment, see Figure 2.

### 4.1.2 Declustering

Probabilistic seismic hazard assessment generally assumes that the earthquakes included in the assessment occur independently

of each other, following a Poisson process. In reality, many events occur as a consequence of other events, such as foreshocks, aftershocks and swarm events. Separating the catalogue into dependent and independent events is referred to as declustering, and numerous techniques exist for this purpose; see review in van Stiphout et al. (e.g. 2012). As discussed in detail in Uski et al. (2015), most declustering methods are designed for plate boundary seismicity, with large events and significant Omori-type aftershock activity. In intraplate Fennoscandia, on the contrary, events are generally small and even moderately large

events can have few and small aftershocks, inconsistent with both the Omori and Båth aftershock relationships. Following Uski et al. (2015), we implemented a declustering scheme using a Gardner and Knopoff (1974) windowing procedure based on conservative estimates of location error and manual inspection of clusters. We set the spatial (radius) and temporal distances between an event and its potential aftershock/swarm member to 10 km and 30 days for events with magnitude larger than 1.5, and to 5 km and 15 days for smaller events. We did not consider the magnitude of completeness or the relative size of events

in the clustering. The largest event in each cluster was considered as the independent event and forwarded to the declustered catalogue.

The results were checked manually and also by comparing inter-event times of subsets of the declustered catalogue with synthetic Poissonian catalogues, produced using the observed average rates and the average of 1,000 simulated inter-event time sequences. The subsets were defined in time intervals where the average yearly rate is approximate similar and visual

inspection of the subset inter-event time distributions (Figure A1) shows that the declustering process to a large degree removes the non-Poissonian short inter-event time events in the original catalogue.

In agreement with Mäntyniemi (1996) and Uski et al. (2015), we find that the fraction of dependent events in Fennoscandia is low. Our catalogue retains 19,943 independent events, implying approximately 18% dependent events. The results show that clustering in Sweden is highest along the very active post-glacial faults in the north, especially the Pärvie and Burträsk

faults, and is also significant along the northeast coast and in some areas south of Lake Vänern (Figure A2). The largest cluster contains 199 events and occurred in the Kouvola area in southern Finland, known for its shallow swarm activity. Uski et al. (2015) found a smaller fraction of dependent events of only 11%, a difference to our result which is likely due to the fact that their area of interest did not include the highly active west coast and Nordland regions in Norway, where swarm activity is





frequently observed (e.g. Shiddiqi et al., 2022). We note that the ESHM20 earthquake catalogue contains a total of 360 events
for the area covered by our declustered catalogue, all with $M_W \geq 3.5$.

### 4.1.3 Magnitude homogenization

Seismic hazard assessments should be based on homogeneous earthquake size estimates, as commonly represented by the
moment magnitude, $M_W$, scale (Hanks and Kanamori, 1979), which does not saturate at large magnitudes. Most earthquakes
in the FENCAT catalogue, however, have magnitude estimates either from macroseismic observations or from local magnitude
scales, $M_L$, with scales varying slightly over time and institute. Uski et al. (2015) constructed a consistent moment based
magnitude scale based on the local Helsinki magnitude scale, $M_L(HEL)$, earthquakes in SNSN after 2000-08-15 which all
have estimates of the scalar seismic moment (Rögnvaldsson and Slunga, 1993) and other moment estimates from individual
studies. They provided relationships between various magnitude scales and $M_L(HEL)$, and related $M_L(HEL)$ to the scalar
seismic moment $M_0$ as

$$M_L(HEL) = 0.83 \log_{10}(M_0) - 7.98 \qquad \text{for } \log_{10}(M_0) \leq 13.5$$
$$M_L(HEL) = 0.59 \log_{10}(M_0) - 4.73 \qquad \text{for } \log_{10}(M_0) > 13.5$$

Uski et al. (2015) points out that the relation is only defined by observations to $\log_{10}(M_0) \approx 17.5$ ($M_W \approx 5.6$), but that it
agrees well with the original Hanks and Kanamori (1979) to approximately $\log_{10}(M_0) = 19$, or $M_W = 6.6$.

The final declustered and magnitude homogenized catalogue has 19,943 events, with magnitudes ranging between $-1.4 \leq$
$M_W \leq 5.9$. There are 341 events with $M_W \geq 3.5$ and 182 events with $M_W \geq 4.0$ in the tectonic source zones, indicating that
our magnitude homogenization scheme produces slightly lower magnitudes than the one used by ESHM20 for Fennoscandia,
which has 360 events with $M_W \geq 3.5$ in the same area. We will only use events recorded from the year 1875 onwards, which
is approximately when the Swedish Geological Society started more systematic investigations of earthquake reports (Kjellén,
1910). Figure 3 shows earthquake magnitude-density plotted as a function of time for the Swedish economic zone and high-
lights the significant improvement in completeness after the year 2000. We also note the lack of reported $M_W \geq 3.2$ events
between 1936 and 1962, the reason for which we do not know, and the very few recorded events with $M_W \geq 4$. Prior to
1950 the data stems almost entirely from macroseismic observations, Figure 3 shows that this has produced a lower magnitude
threshold at approximately $M_W$ 2.

### 4.2 Seismic source areas

285 Seismic source areas (SSAs) are defined based on geological and tectonic features, the clustering of seismicity and homogeneity
of the seismicity rate. In addition, the observation of earthquake activity in the zone should be homogeneous over time such
that variations of the magnitude of completeness within each zone are the same. These source areas form the basis upon
which recurrence parameters, maximum magnitude and ground motion models are calculated and assigned. During the Finnish
nuclear industry projects (Korja and Kosonen, 2015; Korja and Kihlman, 2016), Fennoscandian seismologists and geologists
290 came together to define appropriate cross-border SSAs for these site-specific PSHAs. Work on harmonized Fennoscandian





SSAs was continued as part of the review for ESHM20 and further developed in workshops in 2022 and 2023 for this study and ongoing studies in Finland and Norway.

As this study focuses on Sweden, we use earthquakes in the area of Sweden and a 300 km wide area surrounding the national boundaries and, in the sea, the Swedish Economic Zone. Following the concept of super zones in ESHM20, we define both large scale Tectonic Source Zones (TSZ) and Area Source Zones (ASZ), see Figure 2. The TSZs capture large-scale aspects of the crust, such as the regional orogenies and deformation structures, while attempting to ensure homogeneous seismicity patterns. Given the relatively large number of earthquakes in the TSZs, they provide large scale recurrence parameters which we use for ASZs with insufficient number of events. We define four TSZs: Tectonic source zone T1 comprises the Archean and Svecokarelian rocks of eastern and northeastern Sweden and northern Finland, including the seismicity along the Swedish northeast coast, most of the Fennoscandian post-glacial faults, the Kuusamo seismicity in Finland and the scattered seismicity in southeastern Sweden. Zone T2 spans the Sveconorwegian rocks in southwestern Sweden and southern Norway plus the Oslo graben, which are all areas of relatively high seismicity. Western and northern Norway with the Caledonides and the adjacent ocean areas make up zone T3, including the intense seismicity along the west coast and in the Nordland region. T4 is the least seismically active area, with the Svecokarelian and Archean rocks of central and southern Finland, and the Baltic countries on the sediment covered East European Platform.

The smaller zones, ASZs, are delineated based on active deformational features, such as post-glacial faults, smaller scale geological and tectonic features and distinct seismicity patterns, see Figure 2. For zones that extend outside the 300 km limit, we include all events in the zones in the calculations in order to increase the significance of the results.

### 4.3 Maximum magnitude

An estimate of the maximum possible earthquake magnitude, $M_{max}$, in each seismic zone is used both when calculating the recurrence parameters using a doubly truncated Gutenberg-Richter distribution (e.g. Weichert, 1980) and when performing the actual hazard calculation. In stable continental regions like Fennoscandia, estimation of $M_{max}$ is complicated by low seismicity rates and short observation times, lack of well-defined active fault zones and general uncertainty about the length, or existence, of an earthquake cycle (e.g. Wheeler, 2016; Calais et al., 2016). A number of different approaches have therefore been taken in assessing $M_{max}$ for PSHA purposes. Two frequently used methods are extreme value type statistics (EVTS) based on observational data (e.g. Kijko and Graham, 1998; Kijko and Singh, 2011; Beirlant et al., 2019; Zöller, 2022) and the so called EPRI method of combining global tectonically analogous datasets with local observations (Johnston et al., 1994; Wheeler, 2016).

The largest observed earthquake in Fennoscandia is the Lurøy, Norway event of 1819, with a magnitude that was revised by Mäntyniemi et al. (2020) from 5.8 to 5.9. Using two varieties of EVTS, Mäntyniemi et al. (2001) found Fennoscandian $M_{max}$ of 5.84±0.50 and 5.94±0.52, close to the observed maximum magnitude (excluding the post-glacial faults). As noted by Kijko and Singh (2011), the EVTS methods often underestimate $M_{max}$. Wahlström and Grünthal (2001) used the EPRI approach and found five point probability distributions of $M_{max}$ for onland Fennoscandia, from $M_{max}$ 5.85 to 7.70, with equal probabilities





of 0.2. ESHM13 and ESHM20 both use the same EPRI based approach to $M_{max}$ for Fennoscandia, with a resulting distribution

of $M_{max}$ of 6.3, 6.6 and 6.9, with probabilities 0.5, 0.4 and 0.1, respectively.

The post-glacial faults of northern Fennoscandia are still seismically active (e.g. Lindblom et al., 2015), although the main ruptures are inferred to have occurred at the end of the latest deglaciation some 10,000 years ago (e.g. Smith et al., 2021). They have surface lengths of 40 - 150 km (Smith et al., 2021), indicating potential magnitudes of up to $M_W$ 8 (Lindblom et al., 2015). Such large events have been inferred to be triggered by the glacial isostatic stress contribution from the late deglacial

phase (e.g. Lund, 2015), but as the recent trenching of the Stuoragurra fault in northern Norway indicate ruptures of magnitude 7 as late as 700 years ago (Olesen et al., 2021), the potential for very large earthquakes in the current stress field needs to be considered.

Here we combine the EPRI approach of ESHM20 with the post-glacial fault information and use an $M_{max}$ distribution of 6.3, 6.6, 7.0 and 7.5, with probabilities 0.4, 0.4, 0.15 and 0.05, respectively. The large magnitudes mostly affect the hazard on

long time scales, as discussed further in Section 6.

### 4.4 Calculating recurrence parameters

Recurrence parameters for each source zone are calculated assuming that the mainshocks in the catalogue follow a Poisson distribution that can be represented by the Gutenberg-Richter (Gutenberg and Richter, 1944) relationship

$$\log_{10} N(M) = a - bM \tag{1}$$

where $N(M)$ is the cumulative number of earthquakes per year with (moment) magnitude greater or equal to $M$, $a$ is the activity rate and $b$ represents the ratio of low-magnitude events to high magnitude events. The relationship breaks down for events with magnitudes smaller than some completeness magnitude $M_c$, where not all events are recorded. Estimating recurrence parameters is challenging in a low seismicity area like Fennoscandia, where larger earthquakes are rare and population density low, such that there are very few recorded events prior to the installation of more sensitive seismic networks in the

1960s and 1970s. Although the first seismograph in Fennoscandia was installed in Sweden in 1904, the completeness magnitude of the catalogue has varied during the 20th century from about M4 to about M2. As Figure 3 shows, due to the rapid expansion of the seismic network in the early 2000, the completeness magnitude for Sweden with the seismic network has dropped to below M1. However, once the catalogue is subdivided into tectonic or area source zones, the number of earthquakes and $M_c$ vary significantly with time. Therefore, a time-varying completeness algorithm (Weichert, 1980), as implemented by

the OpenQuake (OQ) engine (Pagani et al., 2014), was used to estimate the recurrence parameters. For quality control, we also used Aki (1965)'s maximum likelihood method, as modified by Tinti and Mulargia (1987). Recurrence parameters, with associated uncertainties, are estimated for the four tectonic source zones and for area source zones with sufficient data.

As is well established, calculation of a- and b-values with small sample sizes or a small magnitude range may lead to significant bias in the estimates, outside the formal uncertainty limits (e.g. Geffers et al., 2022). We therefore require that a

zone contains more than 200 events for a recurrence parameter calculation. For area source zones sparse in data we follow the ESHM20 methodology and assign them the b-value from the corresponding tectonic source zone, while an a-value is





calculated by re-scaling the tectonic zone a-value by the ratio of the number of events with magnitude above $M_c$ in the two zones, $N_{ASZ}/N_{TSZ}$. If there are no events in the zone, the tectonic a-value is scaled by the areal ratio of the two zones.

In most zones we use data from the late 1800 or early 1900 onward. In the far north, however, where there is very little macroseismic data, our records start with the increase in station density in the 1960s. The number of complete events in each tectonic source zone range from 262 in T4 to 7347 in T1. The a-values range from 1.857 in T4 to 3.267 in T3, and the b-values from 0.842 in T2 to 0.912 in T1, see Table 1. In the ASZs, recurrence parameters could only be calculated for 12 out of the 31 zones (zones 2, 3, 4, 9, 13, 14, 15, 18, 23, 24, 30 and 31) due to the sparsity of data in most zones. For zones within 300 km of Sweden, the a-values for the ASZs range between 0.041 for zone 20 in T4 to 2.504 for zone 2 in T1. The b-values range between 0.8 for zone 18 in T1, to 1.051 in ASZ 2, see Table 2. Uncertainties are calculated using the OQ implementation of the Weichert algorithm. Zone 23 has the largest b-value uncertainty of about 5%, followed by zone 4 at around 4%. Due to the large number of events with magnitude above $M_c$ in tectonic zone T1, the b-value uncertainty is only $\pm 0.007$, which we find unreasonably small compared to how the b-value changes when $M_c$ is varied. We therefore set the minimum b-value uncertainty to $\pm 0.01$. In zones with too few events for a separate calculation of recurrence parameters, we assign the same uncertainty as what we find for the tectonic zone calculations to the a- and b-values. As the a-value is scaled by the number of complete events, this implies that the a-value uncertainty becomes relatively very large in some ASZs.

### 4.5 Source model logic tree

The source model logic tree is based on the earthquake catalogue, the zonation, the derived recurrence parameters for each zone and their uncertainties, and the maximum magnitude information, see Figure 5. The first branching level represents each individual area source zone, followed by three branches representing the maximum likelihood estimate of the doubly truncated GR distribution and two standard deviations, representing the 5th and 95th percentiles. Four further branching levels are applied to each model for the $M_{max}$ values of 6.3, 6.6, 7.0 and 7.5, each with 0.4, 0.4, 0.15 and 0.05 as their respective weights. The overall structure of the source model logic tree is as implemented in the ESHM20 calculations for the region (Danciu et al., 2021). With three branches each for the a-, b- values and four for $M_{max}$, a total of twelve branches are defined, Figure 5.

### 4.6 Ground Motion Models

The attenuation characteristics of the crust are of utmost importance for a seismic hazard assessment as they determine how the ground motions from an earthquake decay with distance. In stable continental regions like Fennoscandia, such ground motion models are difficult to construct as the low seismicity rate of the region, the region's size and the relatively sparse seismic networks imply that there is usually very little strong motion data available. Notwithstanding, Fülöp et al. (2020) used weak motion data from events in Finland and Sweden between 2006 and 2018, with magnitudes $1.5 \leq M_L \leq 4.2$, in the development of a Fennoscandian Ground Motion Model (GMM), Fenno-G16. In order to make up for the lack of large events, they added data from an eastern Canadian subset of the NGA-East data (Goulet et al., 2014), with largest magnitude $M_W$ 6.76, and implemented Fenno-G16 based on an adjustment of the backbone curves of the G16 equation developed by Graizer (2016)


for Central and Eastern North America (CENA). The released version of Fenno-G16 contains only one branch in the logic tree
and uses a $V_{S,30}$ for very hard rock of 2800 m/s.

The recently developed scaled backbone GMM for cratonic regions in ESHM20 (Weatherill and Cotton, 2020), described in
Section 3.1, also partly incorporated NGA-East data and models (Goulet et al., 2018) and used the Fülöp et al. (2020) data set
to aid in deciding the weights for the two main branches of cratonic (0.8) and shallow crustal (0.2) in the logic tree.

In this study, we use the two recently developed cratonic GMMs Fenno-G16 (Fülöp et al., 2020) and ESHM20 cratonic
(Weatherill and Cotton, 2020) together with the shallow crustal seismicity model used for much of Europe in ESHM20 (Kotha
et al., 2020). In order to study the effects of the different GMMs on the hazard estimates, we use our earthquake data set and
seismic source zones and calculate hazard at a specific site for each of the following logic tree combinations:

1. The native ESHM20 logic trees for shallow-default and cratonic regions, for a $V_{S,30}$ = 800 m/s, which is the ESHM20
   default.

2. The native ESHM20 logic tree for shallow-default regions and the ESHM20 cratonic logic tree with $V_{S,30}$ set to 3 km/s.

3. The ESHM20 logic tree for the shallow-default regions and the single-branch Fenno-G16 GMM for the cratonic regions.
   $V_{S,30}$ = 3 km/s for basement sites and $V_{S,30}$ = 800 m/s for the sedimentary rocks in Skåne, Öland and Gotland.

4. The ESHM20 logic tree for the shallow-default regions and for the cratonic regions two branches, one with a modified
   ESHM20 cratonic logic tree where the 20% branch with the shallow crustal GMM has been removed, weight 0.6, and one
   with the single-branch Fenno-G16, weight 0.4, see Figure 6. $V_{S,30}$ = 3 km/s for the cratonic regions. Figure 6 shows the
   ESHM20 five branch implementation for epistemic uncertainty of the GMM, $\sigma_\mu$ (Weatherill and Cotton, 2020), where
   the $\epsilon_{ii}$ and weights assigned to each branch are determined by a discrete approximation of the Gaussian distribution
   depending on the number of standard deviations to include (Danciu et al., 2021; Miller and Rice, 1983). The epistemic
   uncertainty of the site amplification factor, $\sigma_{\mu,S}$, is similarly included with three branches, however, as we do not adjust
   the site amplification to $V_{S,30}$ = 800 m/s for cratonic areas these branches cancel.

Figure 7 shows how the hazard curves at the location of Uppsala (Fig. 1) vary with the different GMM combinations. The
mean hazard estimates are rather similar, we see that the Fenno-G16 model consistently produces slightly higher PGAs at the
same probability of exceedance (PoE) than the ESHM20 models and that using a $V_{S,30}$ of 3 km/s consistently lower the hazard
compared to a $V_{S,30}$ of 800 m/s, as expected. The uncertainty distribution for Fenno-G16 is generally narrower than those for
the ESHM20 models, for which the lower 0.16 fractile limit extends to significantly lower PGA for the same PoE. Model 5,
with the combination of Fenno-G16 and ESHM20, has a mean hazard similar to Fenno-G16 at higher PoE but moves toward
the midway point between the two at higher PoE. The 0.16 and 0.84 fractile distributions behave similarly.

We choose model 5 as our preferred ground motion model for cratonic areas in this study, as it combines the two most
recent cratonic GMMs, incorporates data and models from eastern North America and takes into account seismic observations
in Fennoscandia. The weights between Fenno-G16 and the ESHM20 GMMs were determined such that we obtain a larger
epistemic uncertainty distribution than Fenno-G16 while somewhat reducing the ESHM20 distribution to lower hazard.


Referring to the zone numbers in Figure 2, we use our cratonic model 5 ground motion logic tree for tectonic zones T1 and T4, encompassing much of Sweden, Finland and the Baltic countries. In tectonic zones T2 and T3 we use the ESHM20 shallow-default logic tree, except for in area source zones 24, 28 and 30, which all use the cratonic combination. As described

in Section 2, Swedish surface rock mostly consists of Proterozoic bedrock which in places is covered by a few meters to tens of meters of glacial sediments. There have been no large scale studies of $V_{S,30}$ in Sweden but generally speaking, many single houses and larger constructions are built on bedrock or using piling to bedrock. We therefore use a $V_{S,30}$ of 3 km/s for all cratonic zones, similar to the 2800 m/s used by Fülöp et al. (2020) and corroborated by e.g. Sadeghisorkhani et al. (2020). The two Baltic islands of Öland and Gotland are covered in sand- and limestone (Rosberg and Erlström, 2019), for these areas we

use the ESHM20 default of $V_{S,30}$ = 800 m/s. In all zones with the shallow-default GMM we also use the default ESHM20 $V_{S,30}$ = 800 m/s.

### 4.7 Definition of OpenQuake parameters

We use the OpenQuake(OQ) engine (v3.12) (Pagani et al., 2014) to estimate the yearly probability of exceedance of peak ground acceleration at the 0.0021 and 0.0004 levels, corresponding to return periods of 475 and 2500 years. The OQ-engine

uses the classical PSHA approach based on the methodology proposed by Cornell (1968) and McGuire (1976). The engine allows for flexibility in modelling seismic sources, using predefined GMMs or implementing new GMMs, and characterizes epistemic uncertainty through a logic tree. The engine incorporates the OpenSHA (Field et al., 2003) calculation structures and workflows, which we use on an equally spaced 5 km calculation grid.

Area sources were defined as per the OQ NRML schema for each ASZ (Schorlemmer et al., 2011). We follow ESHM20 and

define the minimum magnitude used for the hazard computation, $M_{min}$, as 4.5 for all sources, as that is where damages may start to appear. We define a depth distribution where ruptures take place, using 5, 10, 20 and 30 km depth weighted as 0.15, 0.35, 0.35 and 0.15. This accounts for the fact that although most events occur down to about 20 km depth, some areas have significantly deeper earthquakes (e.g. Veikkolainen et al., 2017). Point ruptures are used as we include no specific faults in the model and for the added benefit of reduced calculation times. Truncated Gutenberg-Richter distributions are defined by the

lower and upper magnitude bounds $M_{min}$ and $M_{max}$, and the previously calculated a- and b- values with their uncertainties in different branches of the logic tree.

The calculation parameters for the OQ engine are summarized as follows: twelve end-branches for area source models of the shallow crust and cratonic regions; 18,072 equally spaced grid points (5 km spacing) defined on land within Swedish land borders; a $V_{S,30}$ = 3 km/s; point ruptures; a weighted depth distribution; predefined nodal plane orientations, vertical plane

striking north with a rake of 0 or 180; 25 intensity measures that cover PGA ground motion discretization levels between 0.005 and 3 g; and calculations for the complete enumerated logic tree.


## 5 Results

We present ground motion exceedance levels for mainshock earthquakes in terms of peak ground acceleration (PGA), using our source model and ground motion model logic tree implementations as discussed in Sections 4.5 and 4.6. PSHA results are

commonly presented as hazard maps showing PGA with a 10% probability of exceedance (PoE) in 50 years, as that level is used for building codes, e.g. in Eurocode 8. A 10% exceedance level in 50 years is equivalent to a statistical (Poissonian) return period of 475 years. ESHM20 uses annual PoEs, where $2.1 \cdot 10^{-3}$ is equivalent to 10% in 50 years and $4 \cdot 10^{-4}$ equivalent to the previously used 2% in 50 years, slightly altering the return period from 2475 to 2500 years. We follow ESHM20 and present our results as hazard maps for return periods of 475 and 2500 years, and as hazard curves for four seismically active regions in

Sweden.

### 5.1 Seismic hazard map for Sweden

Seismic hazard maps for Sweden with 475 and 2500 year return times are shown in Figure 8. As usual in probabilistic seismic hazard assessment, hazard closely follows the observed earthquake distribution. The maps show the mean PGA, and we see that for 475 year return period the highest hazard is estimated in the northernmost part of Sweden, in the area of the post-glacial

faults and at the northern shores of the Bay of Bothnia, where hazard reaches 0.05g. Much of the northeastern coast of Sweden shows a mean PGA of 0.03 - 0.04g and the southwest of Sweden shows a mean PGA 0.02 - 0.03g. The southeast and northwest of Sweden have an estimated hazard of less than 0.02g. For the 2500 year hazard map, hazard is similarly distributed, but the highest mean PGA is now 0.15g in the northeast of Sweden, followed by the northeast coast at about 0.1 - 0.125g and the southwest of Sweden between 0.075 - 0.1g. As before, the southeastern and the northwestern regions show the lowest hazard

of less than 0.05g.

We note that the earthquake activity in the Oslo graben in Norway, and even the activity on the Norwegian west coast, affect the hazard levels in southwestern Sweden to some degree. Interestingly, we do not see the same influence from the Nordland region in Norway, where seismic activity is high and where the largest known earthquake in Fennoscandia occurred in 1819. This is likely due to the relatively high b-value we estimate for ASZ 4, and the fact that many of the events occur in swarms

which are removed in the declustering.

### 5.2 Hazard curves for seismogenic areas

The regions in Sweden with the highest seismic activity can broadly be defined as: the southwest around Lake Vänern, the northeast coast and the post-glacial fault province in the north. In Figure 9 we illustrate the seismic hazard, using hazard curves with the annual probability of exceedance versus PGA, at four locations in these regions: the post-glacial Pärvie and Burträsk

faults, the Hälsingland area around Hudiksvall and Västergötland south of Lake Vänern, see stars in Figure 1. We see that the hazard curves are similar for all four regions, the Pärvie and Burträsk faults having slightly higher short term PGA estimates and Västergötland the lowest hazard overall, although all four are within the uncertainty limits of each other. We restrict our curves to a lower limit of an annual PoE of $10^{-5}$, with the 0.16 and 0.84 fractile estimates ranging between 0.3g and 0.9g at this





limit. Comparing to the Uppsala curves in Figure 7, we note that the short term hazard is similar in Uppsala and Västergötland,
but that the longer term hazard is significantly lower in Uppsala.

## 6   Discussion

The result of our seismic hazard assessment, as illustrated in the maps in Figure 8, differ from many of the previous assessments
in that the estimated hazard is largest in northernmost Sweden. This is due to the new emerging understanding of the seismic
activity on the post-glacial faults (PGFs) in northernmost Fennoscandia (Lindblom et al., 2015). The expansion of the Swedish
national seismic network in the early 2000s, from 2 to 22 stations north of 63.5° latitude (Lund et al., 2021), has enabled
observations of previously unknown seismic activity rates on these faults (Lindblom et al., 2015). Almost 3700 earthquakes
have been observed in the region in the last two decades (Figure 4), all but two have magnitudes below 3 which is consistent
with the few observations prior to the network expansion. The very low population density and limited reporting possibilities
in earlier times probably explains the almost complete lack of observations in e.g. the Pärvie area prior to 1967. Unfortunately,
there are no conclusive data on current slip rates on the PGFs. For example, the InSAR study by Mantovani and Scherneck
(2013) inferred vertical displacements at a few segments along the Pärvie fault, but did not conclude any slip rates. This
complicates the inclusion of the post-glacial faults explicitly in a PSHA.

The inference that the last 20 years of earthquake data in northern Sweden indicates that the area has high seismic hazard is
interesting in the light of recent work on the PGFs in the region. Smith et al. (2021) found repeated ruptures on the Lainio PGF,
with the latest recurring after deglaciation, and (Olesen et al., 2021) identified ruptures of magnitude 7 as recently as 700 years
ago on the Stuoragurra fault in northern Norway. Taken together, all observations suggest that seismic hazard should be taken
into careful consideration in the region, especially since the region is home to major mining operations, large hydroelectric
power dams and an increasing number of wind energy installations.

### 6.1   Estimation of recurrence parameters

With Fennoscandia being a low-seismicity region, estimating the recurrence parameters is a major challenge in the seismic
hazard assessment process. The problem is multifaceted and involves the low seismicity rate, the few large events and the
magnitude homogenization process. As the rate of earthquakes large enough to be felt is low (Lund et al., 2021), there is little
data prior to the installation of an improved seismic network in the 1960s. That network consisted of only six analogue stations,
so the more significant increase in earthquake observations came with the modern SNSN, from the year 2000 onward, when
the completeness magnitude fell below M1 (Figure 3) for a large part of the country (Lund et al., 2021). Many of the observed
larger events occurred in earlier times and therefore have macroseismic magnitudes with significant uncertainty. Later events
have a variety of local magnitudes, which complicates the magnitude homogenization procedure (Uski et al., 2015). In addition,
the magnitude estimation for the many small events in our dataset comes with appreciable uncertainty, e.g. due to low station
density, which further complicates the magnitude homogenization procedure.





As discussed in Section 4.4, we use the Weichert (1980) method, as implemented by OpenQuake, to calculate the recurrence parameters, using data reaching back to 1875. We also use Aki's method (Aki, 1965), as revised by Tinti and Mulargia (1987), with the post-2000 data to add confidence in the results. We found that varying $M_{max}$ during parameter estimation has no significant effect on the results. However, the results are sensitive to variations in the magnitude of completeness, a change in $M_c$ by 0.2 can lead to a significant change in a- or b-value, beyond the formal uncertainties of the original results. The
magnitude of completeness is non-trivial to estimate, manually or using some curvature method, especially for the very sparse historical data. It is thus likely that uncertainties in a- and b-values are sometimes larger than the formal estimates, which led us to set a minimum b-value uncertainty to 0.01. This was needed for tectonic zone T1, which contains 7347 events above $M_c$ and therefore produces a very small formal uncertainty.

    We note that in their investigation of the sensitivity of site-specific PSHA in Finland to variations in the input parameters,
Fülöp et al. (2023) found that $M_{max}$ variations do not affect the PGA results significantly but is more important at low ground motion frequencies at an annual frequency of exceedance of $10^{-5}$. As expected, they point out that a- and b-value variations have a large effect on the results.

## 6.2   Comparisons with previous seismic hazard studies

As discussed above, the most significant difference between our seismic hazard model and previous studies is the spatial
distribution of hazard in northern Sweden. The Wahlström and Grünthal (2001) study found the largest hazard in Sweden focused in the area surrounding the 1904 $M_W$ 5.4 earthquake on the west coast, with a median PGA of 0.03 - 0.035g for a 475 year return period. This is slightly higher than our estimate of about 0.025g in that area. Similar to us, they found that southeastern and northwestern Sweden, east of the mountain range, has the lowest hazard, below 0.015g. Along the northeast coast they estimate a PGA of 0.015 - 0.02g, lower than our approximately 0.03g, and in the northernmost region of Sweden
their 475 year PGA estimates vary between 0.01 - 0.02g, significantly lower than our 0.03 - 0.05g. We note that Wahlström and Grünthal (2001) use the median hazard for their estimates. As we use mean estimates, our hazard values are likely to be generally somewhat higher.

    The Mäntyniemi et al. (2001) hazard map is similar to the one from Wahlström and Grünthal (2001). They find high hazard in the southwest and along the northeast coast, with the 475 year PGA reaching 0.02 - 0.025g in both regions, similar to our
results. In the north, their hazard estimate is lower than ours, at 0.01 - 0.015g. We note that both (Wahlström and Grünthal, 2001) and (Mäntyniemi et al., 2001) estimate a higher hazard in the northern Swedish mountain region than us, indicating that their area source zone definition is such that the high seismicity in Norwegian Nordland produces significant hazard in Sweden in their model, or that they estimate lower b-values than we do in that region.

    The 2013 European seismic hazard model (ESHM13, Woessner et al., 2015) found the highest hazard in Sweden to be
focused in a narrow region across the southernmost part of the country and up along the west coast, with a maximum PGA for 475 year return period at approximately 0.025g in the south. This is significantly more than our estimate of less than 0.01g in that area. Along the southwest coast of Sweden the hazard in ESHM13 is about 0.02g, similar to ours, while the rest of the country has a hazard between 0.001 - 0.02g, again significantly less hazard in the north than in our model.





The most recent iteration of the European seismic hazard model (ESHM20, Danciu et al., 2021) changed the distribution of
hazard in Sweden significantly as compared to ESHM13. Hazard estimates were lower in the south and southwest and higher
in south-central parts of the country and up the northeast coast, following the coastline around the northern Bay of Bothnia.
This agrees with the distribution of hazard in our model, except that in our model the hazard is lower in the southeast. We
illustrate the differences between our model and ESHM20 for 475 and 2500 years return periods in Figure 10, which shows
that our model produces higher hazard than ESHM20 for both return periods in most the country. Again, the most notable
difference is in the north, where ESHM20 uses the recurrence parameters from the large tectonic zones but where we have very
much more data (Figure 4), such that we can better constrain the a- and b-values. Comparing recurrence parameters we note
that the ESHM20 area source zone NOAS376 in northern Fennoscandia, encompassing many of the PGFs, is approximately
30% larger in size than our ASZs 13, 14 and 15 combined. The ESHM20 b-value is $0.91\pm0.032$ in that area, which is within
the uncertainties of the b-values in our ASZ 13 and 14, and just above the uncertainty limits of ASZ 15, see Table 2. The
ESHM20 a-value $1.584\pm0.0059$ is re-scaled from the tectonic zone and significantly lower, considering the much larger area,
than our a-values around 1.9 - 2.0 (Table 2) for the individual zones 13, 14 and 15. We see in Figure 10 that our PGA is more
than 0.04g higher for a 475 year return period than the ESHM20 estimate in some areas in the north. For a 2500 year return
period the difference is 0.125g. Further south along the northeast coast our hazard is still higher by some 0.02g, a difference
which decreases towards the southeast and along the southeast coast the hazard in our model is smaller than in ESHM20. In
the southwest, our model estimates higher hazard than ESHM20 by about 0.015g. We note that ESHM20 and our model agree
well along the northwestern border, where none of the models see much influence of the Norwegian coastal seismicity.

### 6.3 Seismic hazard and risk in Sweden

Comparing to surrounding regions, the seismic hazard we find in northern Sweden is only surpassed by the hazard in western
Norway around Bergen and in the Rana region in northwestern Norway. Our new results for northern Sweden will be compared
to results obtained in neighbouring countries in order to calibrate the hazard along the borders. Probabilistic seismic hazard
projects are ongoing in both Norway and Finland, and we aim to further the work on a joint homogenized Fennoscandian
seismic hazard model in the near future.

Our results show that the hazard correlates well with the observed seismicity. We note, however, that most of the seismic
activity is of low magnitude and that there is not a clear correlation between the observed larger events, above e.g. $M_W$ 4,
and more intense microearthquake activity. The Tornquist zone in southern Sweden, for example, has experienced five events
with $M_W \geq 3.7$ in the last century but has a low seismicity rate whereas the Burträsk fault, the most seismically active area in
Sweden, has seen no such events during that time. There was a $M_W$ 3.9 event in west-central Sweden in 2014 in an area with
only a few prior, small events (Lund et al., 2014), and the large magnitude 5 events in Kaliningrad in 2004 (Gregersen et al.,
2007) occurred in an area with very few previously observed events. These surprise events, and the sometimes poor correlation
between the rate of small events to large, points to the difficulty of PSHA in stable continental regions. Adding to the problem
is the largely unknown level of temporal and spatial stationarity of earthquakes in these regions.





The large post-glacial faults of northern Fennoscandia pose an interesting problem to seismic hazard assessment. Previously inferred to have ruptured only once, at the time of final deglaciation of the region (e.g. Lagerbäck and Sundh, 2008), the main question concerned whether or not the faults could rupture again without an intervening major glaciation. Glacial isostatic

adjustment modelling (see review in e.g. Lund, 2015) shows that glacially induced stresses combine with the tectonic stress state and bring the PGFs towards instability at deglaciation. Due to the slow strain accumulation in the Fennoscandian Shield, repeated large ruptures on the same faults segments have been considered unlikely with repeat times less than 100,000 years (Korja and Kosonen, 2015). However, as alluded to above, more recent investigations have found that the PGF systems have seen multiple ruptures along the faults and also multiple ruptures at the same location, see reviews for Fennoscandia in Steffen

et al. (2021). Including the PGFs in a PSHA is currently only possible as an increase in $M_{max}$ for area source zones including PGFs, with a weight that is difficult to assess due to the current lack of data. We have seen that this does not have a large effect on the estimated PGA but is more likely to affect the longer periods of ground motion (Fülöp et al., 2023).

A more appropriate mechanism, at this time, to take the PGFs into account is through deterministic seismic hazard assessment (DSHA). The faults are mapped in high resolution (e.g. Smith et al., 2021) and the depth extent of current seismicity is

known for some PGFs (e.g. Lindblom et al., 2015), making it possible to use scaling relations to estimated potential rupture magnitudes. While a deterministic scenario may have a very low probability of occurrence, a DSHA will provide additional insights from a risk reduction perspective and potentially highlight events that may dominate the seismic risk in a certain area or for a specific site, and therefore must be considered (McGuire, 2001). Performing site-specific seismic hazard, and risk, assessment in the post-glacial fault province in northern Fennoscandia for critical infrastructure, such as dams where low annual

probabilities must be considered, a combined PSHA/DSHA approach may be the most appropriate.

## 7    Conclusions

This study provides a probabilistic seismic hazard model of Sweden based on updated earthquake catalogues from the Swedish national seismic network (Lund et al., 2021) and neighbouring networks, and recent results and methods from the European seismic hazard model ESHM20 (Danciu et al., 2021). The hazard is calculated using the OpenQuake engine (Pagani et al.,

2014), using earthquakes since 1875 represented through 31 area sources zones, and recently developed ground motion models (Fülöp et al., 2020; Weatherill and Cotton, 2020; Kotha et al., 2020). A weighted logic tree approach is used to represent the uncertainties in the recurrence parameter estimates, $M_{max}$ and the ground motion models. The resulting hazard estimates are calculated as peak ground-acceleration (PGA) values and mapped in Figure 8 for 475 and 2500 years return periods.

The updated earthquake data allow us to use a large number of smaller events that have not previously been available, or

used, in seismic hazard assessment. This makes it possible to better constrain the recurrence parameters (a- and b-values) in much of the country, especially for the post-glacial fault province in northern Sweden.

We find a seismic hazard distribution in Sweden which is significantly different to that of earlier studies. The new data from the north indicates that hazard in northernmost Sweden is higher than elsewhere in country, both at 475 and 2500 years return period. For a return period of 475 years, mean PGA along the post-glacial faults and the northern coastal tip of the


Bay of Bothnia is estimated to be about 0.04 to 0.05g. For the rest of the north-eastern coast, mean PGA is estimated to be 0.02-0.03g. The lowest PGA estimates in the country, 0.02g and lower, are found in the south-eastern and north-western parts of the country. Similar trends are seen for the 2500 year PGA estimates, with the highest mean PGA of 0.15g seen in the north-east, 0.1-0.125g for the post-glacial faults and the north-eastern coast, 0.075-0.1g for the south-west and less than 0.05g for the south-east and the north-west. Hazard curves for four locations in some the most active areas of the country, the Pärvie

and Burträsk faults, the Hälsingland area and south of Lake Vänern, show similar hazard in these areas, with slightly higher hazard along the post-glacial faults.

   Compared to ESHM20 (Danciu et al., 2021), our model shows somewhat higher hazard in the more seismically active areas of Sweden but slightly lower hazard in the less active areas. The main difference is in the north, where we find significantly higher hazard. We note that our declustered, homogenized earthquake catalogue contains 19,943 events, with magnitudes down

to just below zero, compared to 360 events in the same area for ESHM20, all with $M_W \geq 3.5$.

   Challenges remain in seismic hazard estimation for Fennoscandia. There are significant uncertainties in magnitude homogenization, for the smaller magnitude events compared to the larger, in the estimate of recurrence parameters, as larger events occur infrequently, and in the estimation of ground motions models for the Fennoscandian Shield, as sparse to non-existing data for larger events and short distances implies that both epistemic and aleatory uncertainties are difficult to estimate. On a

larger scale, the occurrence of large events in areas with little prior seismicity and the uncertainties surrounding the potential for large earthquakes on the post-glacial faults in northern Fennoscandia pose problems for seismic hazard and risk assessment, and require continued studies.

*Data availability.*

   The earthquake catalogue was prepared based on data obtained from the Swedish national seismic network (SNSN, 1904),

the Norwegian national seismic network (University of Bergen, 1982), the Finnish national seismic network (FNSN, 1980), The Estonian seismic network (Geological Survey of Estonia (GSE), 1998) and the Geological Survey of Denmark and Greenland (GEUS, 2023).

*Code and data availability.*

   The QGIS (https://www.qgis.org/en/site/) software was used to delineate and visualize the zones. The OpenQuake engine

(https://github.com/gem/oq-engine) and its included suite of data preparation programs were used to calculate recurrence parameters, create input files, and run the hazard calculations themselves. The hazard maps were plotted using the plotting function build into OpenQuake and pygmt, and the hazard curves were plotted using matplotlib.

*Author contributions.*



Niranjan Joshi: Data preparation, processing, investigation and visualization, writing. Björn Lund : Conceptualization,
methodology, supervision, data preparation and analysis, review, editing and writing. Roland Roberts: Conceptualization,
methodology, supervision, review, editing.

*Competing interests.*

The authors declare no competing interests.

*Acknowledgements.*  The authors gratefully acknowledge discussions with Fennoscandian colleagues and members of the ESHM20 team.
We also thank the FENCAT team at the Institute of Seismology, Helsinki University, and the Norwegian National Seismic Network for
earthquake data and advice on magnitudes.



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

**Figure 1.** Seismicity, geology, and tectonic map of Fennoscandia. Black dots indicate earthquakes in the declustered catalogue in this study. Yellow stars indicate locations referred to in the text, from north: the Pärvie fault, the Burträsk fault, Hudiksvall, Uppsala and south of Lake Vänern. The Sorgenfrei-Tornquist and Trans European Suture Zones with dashed black lines. Post-glacial faults from Munier et al. (2020). Map modified after Gregersen et al. (2021)




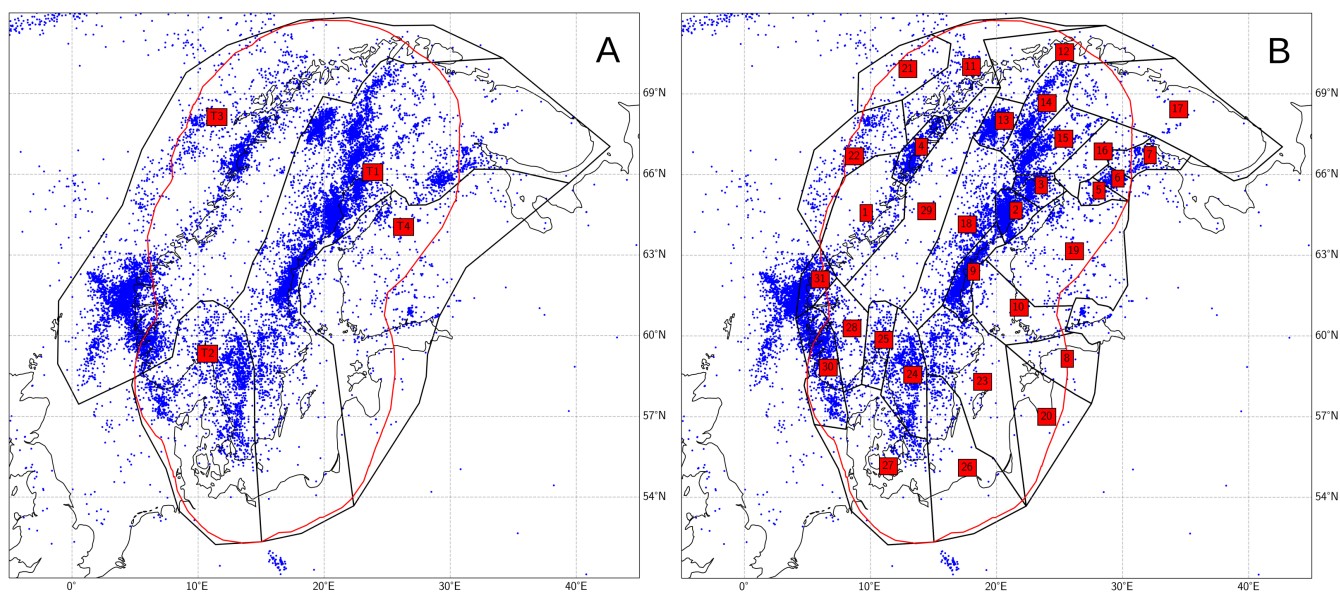

**Figure 2.** Tectonic source zonation scheme (A) and the area source zonation scheme (B) used in this study. The red line indicates a zone encompassing Sweden that is 300 km from the Swedish border or economic zone boundaries. Numbers detail the zone numbers.

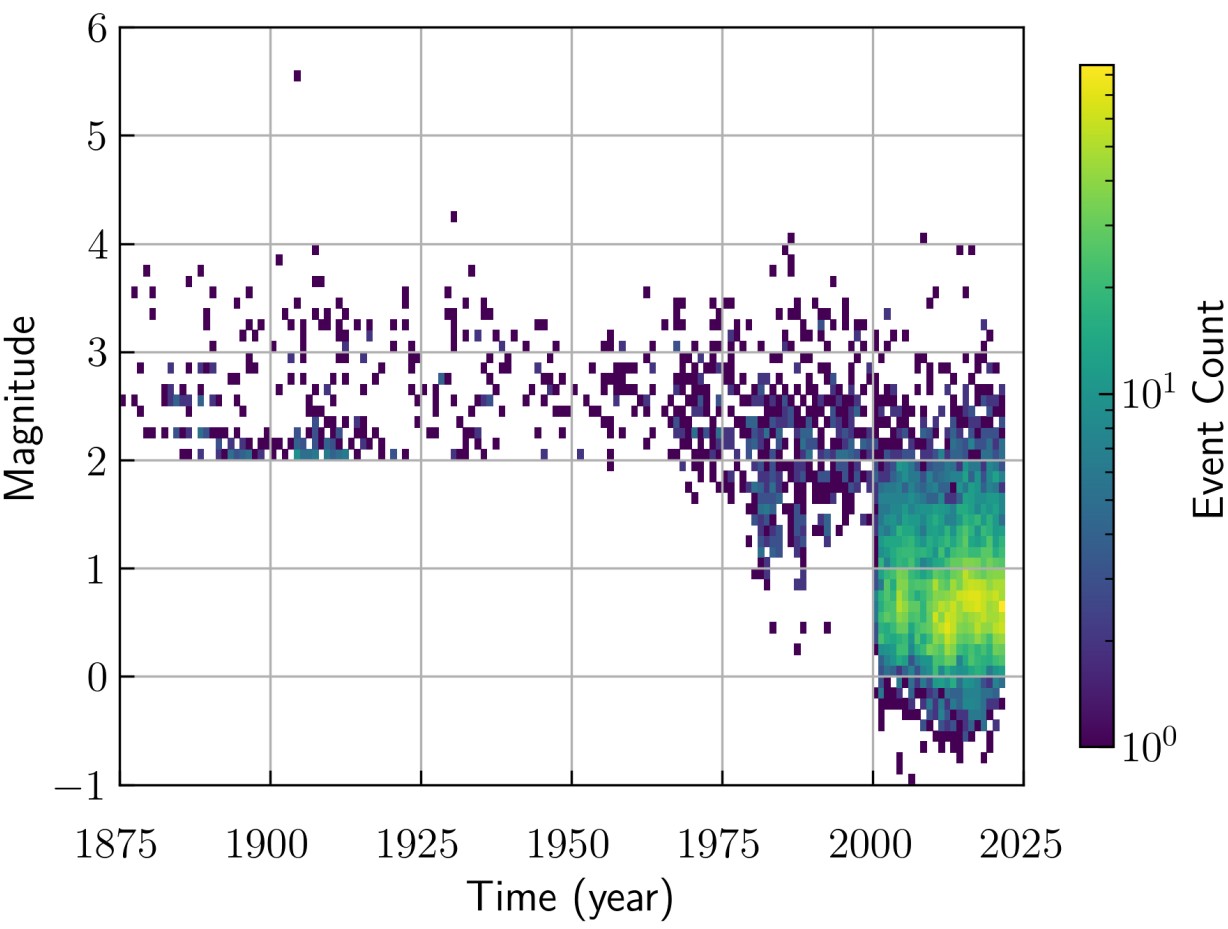

**Figure 3.** Magnitude-time density plot of the earthquakes in the Swedish economic zone, from the declustered and magnitude homogenized catalogue.

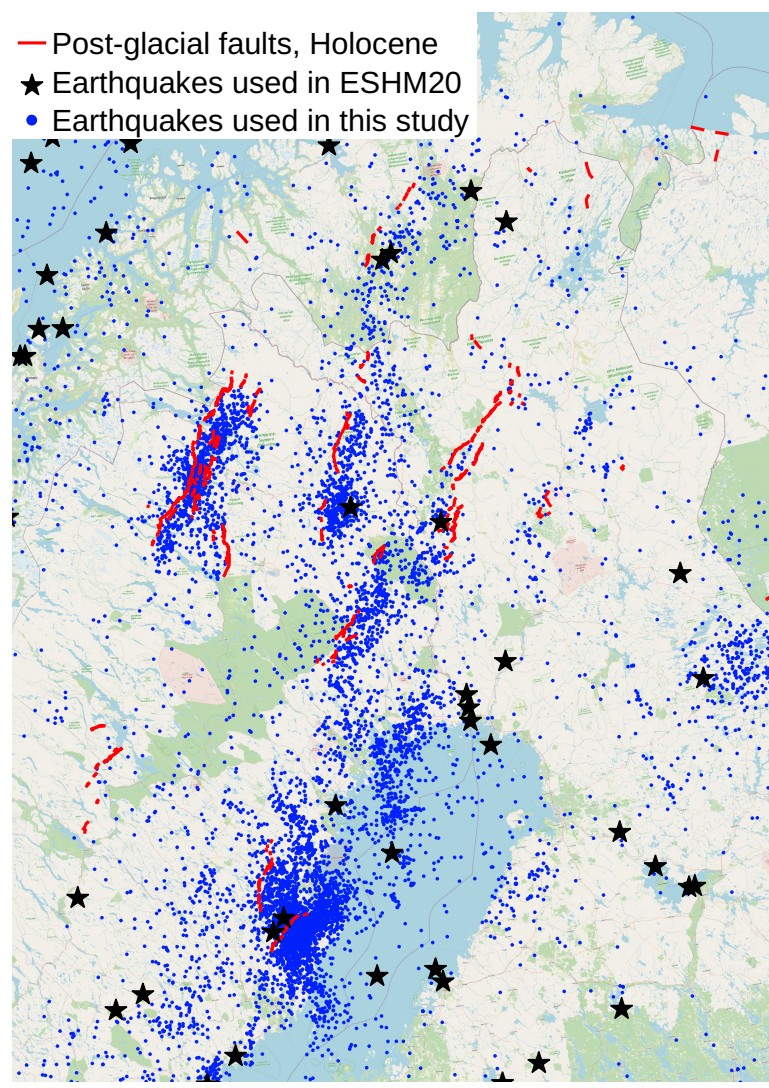

**Figure 4.** Earthquakes in the ESHM20 catalogue (black stars) and our catalogue (blue dots), north of 63 degrees latitude and between 17 and 32.5 degrees east longitude. Red lines: post-glacial faults.




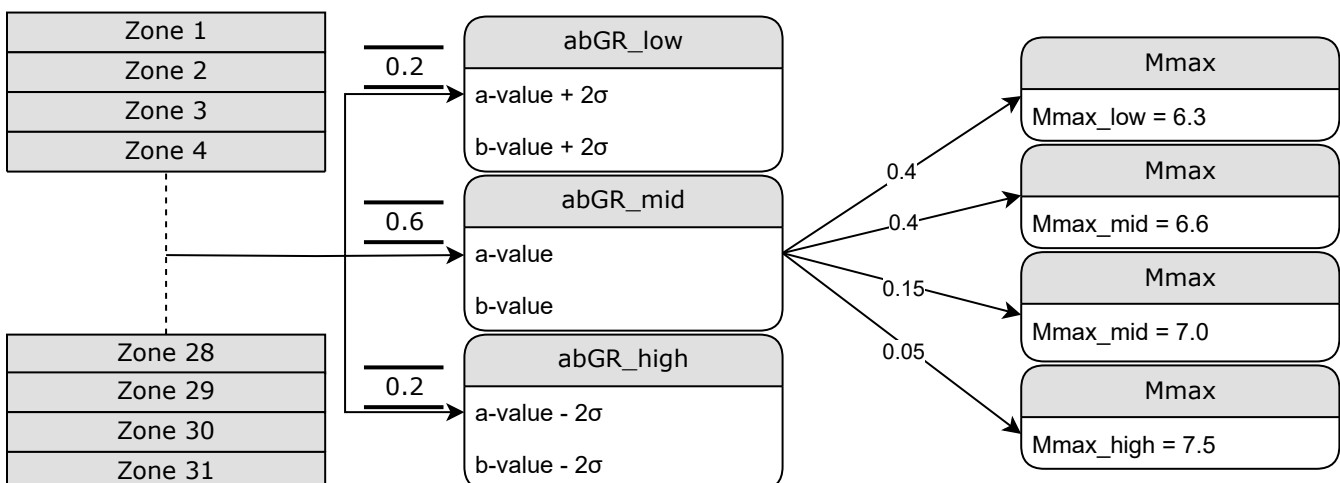

**Figure 5.** Source model logic tree. The numbers indicate the weights of the different branches.





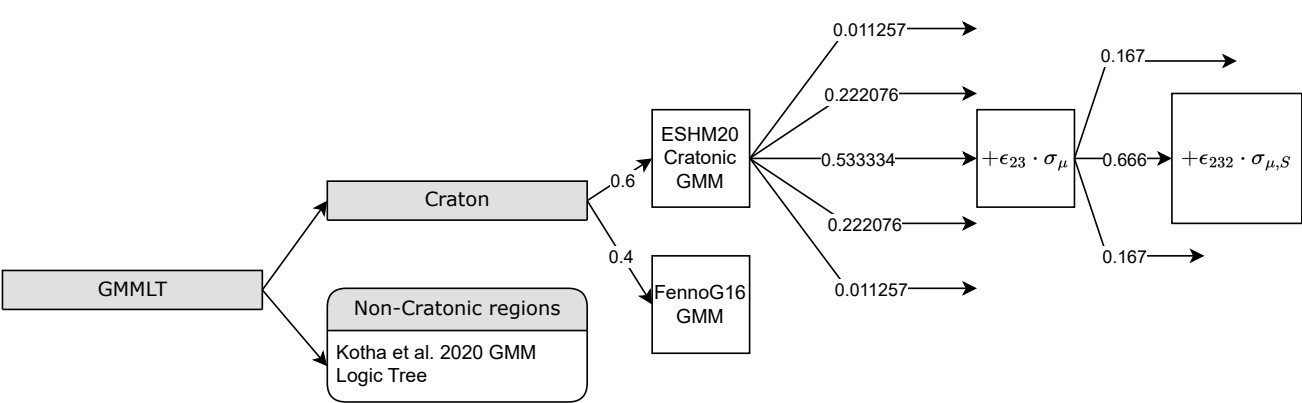

**Figure 6.** Ground motion model logic tree. The numbers indicate the weights of the different branches. The logic tree is further detailed in Section 4.6.



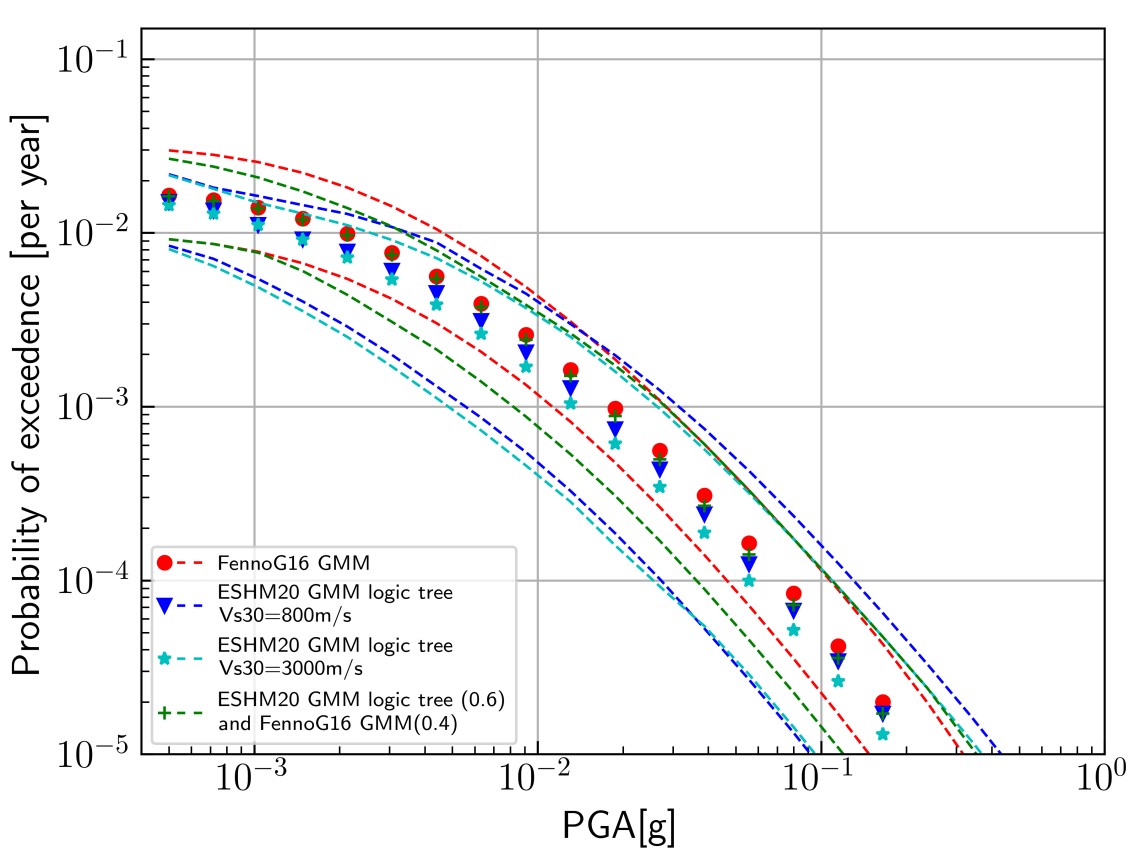

**Figure 7.** Mean hazard curves for Uppsala, Sweden, for different GMM implementations, see Section 4.6 for details. Yearly probability of exceedance versus PGA in g. Solid lines show the mean, while the upper and lower dashed lines show the 0.84 and 0.16 fractiles.



**Figure 8.** Seismic hazard maps for return periods of 475 years (left) and 2500 years (right). Contour lines represent 0.01g for the 475 year map and 0.025g for the 2500 year map. Post-glacial faults indicated by black lines.


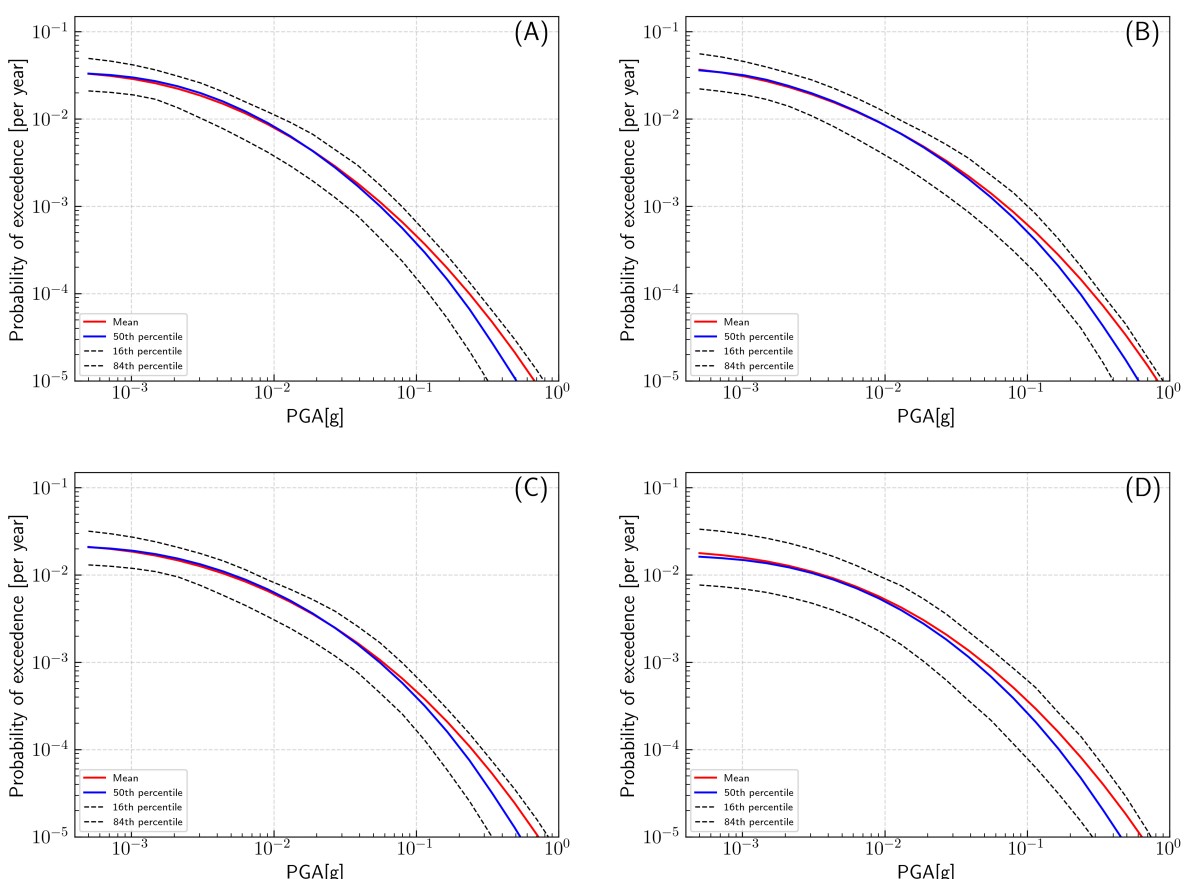

**Figure 9.** Seismic hazard curves for areas of increased seismicity in Sweden (yellow stars in Figure 1). (A) Pärvie, (B) Burträsk, (C) Hälsingland, (D) Västergötland. Annual probability of exceedance versus peak-ground acceleration in g.



**Figure 10.** Seismic hazard maps showing the difference between our and ESHM20 results for return periods of 475 (left) and 2500 (right) years. Contour lines represent 0.01g for the 475 year map and 0.025g for the 2500 year map. Post-glacial faults indicated by black lines.

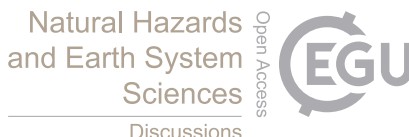

**Table 1.** Recurrence Parameters for Tectonic Source Zonation Schemes.

| Zone | a-value$\pm\Delta a$ | b-value$\pm\Delta b$ |
|---|---|---|
| 1 | 2.926 $\pm$ 0.005 | 0.912 $\pm$0.01 |
| 2 | 2.451 $\pm$ 0.022 | 0.842 $\pm$0.034 |
| 3 | 3.267 $\pm$ 0.013 | 0.895 $\pm$0.023 |
| 4 | 1.857 $\pm$ 0.026 | 0.887 $\pm$0.034 |



**Table 2.** Recurrence Parameters for Area Source Zones in Sweden.

| Zone | a-value$\pm\Delta a$ | b-value$\pm\Delta b$ |
|:---:|:---:|:---:|
| 2 | $2.504 \pm 0.012$ | $1.051 \pm 0.023$ |
| 3 | $1.864 \pm 0.019$ | $0.878 \pm 0.03$ |
| 9 | $2.267 \pm 0.012$ | $0.957 \pm 0.022$ |
| 13 | $2.041 \pm 0.017$ | $0.922 \pm 0.03$ |
| 14 | $2.011 \pm 0.018$ | $0.88 \pm 0.031$ |
| 15 | $1.917 \pm 0.019$ | $0.844 \pm 0.03$ |
| 18 | $1.767 \pm 0.02$ | $0.8 \pm 0.028$ |
| 23 | $1.676 \pm 0.028$ | $0.891 \pm 0.047$ |
| 24 | $2.132 \pm 0.028$ | $0.895 \pm 0.047$ |
| 25 | $1.586 \pm 0.022$ | $0.842 \pm 0.034$ |
| 26 | $1.068 \pm 0.005$ | $0.912 \pm 0.01$ |
| 27 | $1.974 \pm 0.022$ | $0.842 \pm 0.034$ |
| 29 | $1.78 \pm 0.013$ | $0.895 \pm 0.023$ |


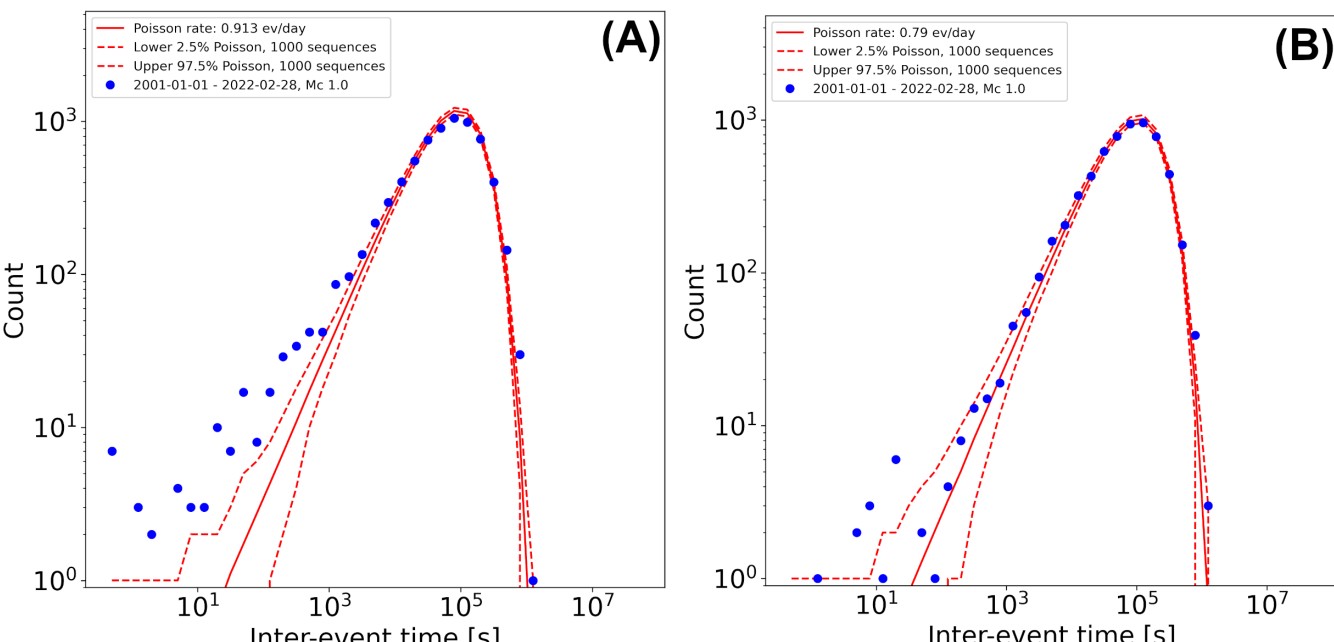

**Figure A1.** Inter-event time distributions for the Fennoscandia earthquake data set from 2001-01-01 until 2022-02-28 and a magnitude of completeness of 1.0. (A) Full catalogue, (B) Declustered catalogue. Data, blue dots. Average Poisson inter-event times, red line. Lower 2.5% and upper 97.5% confidence limits, dashed red lines.




**Figure A2.** Map of Fennoscandia with events belonging to clusters resulting from the declustering algorithm. Cluster sizes indicated by the colours of the dots: 2 events: gray, 3-5 events: blue, 6-10: green, 11 - 20: red, above 20: orange. Larger clusters plotted on top of smaller clusters.


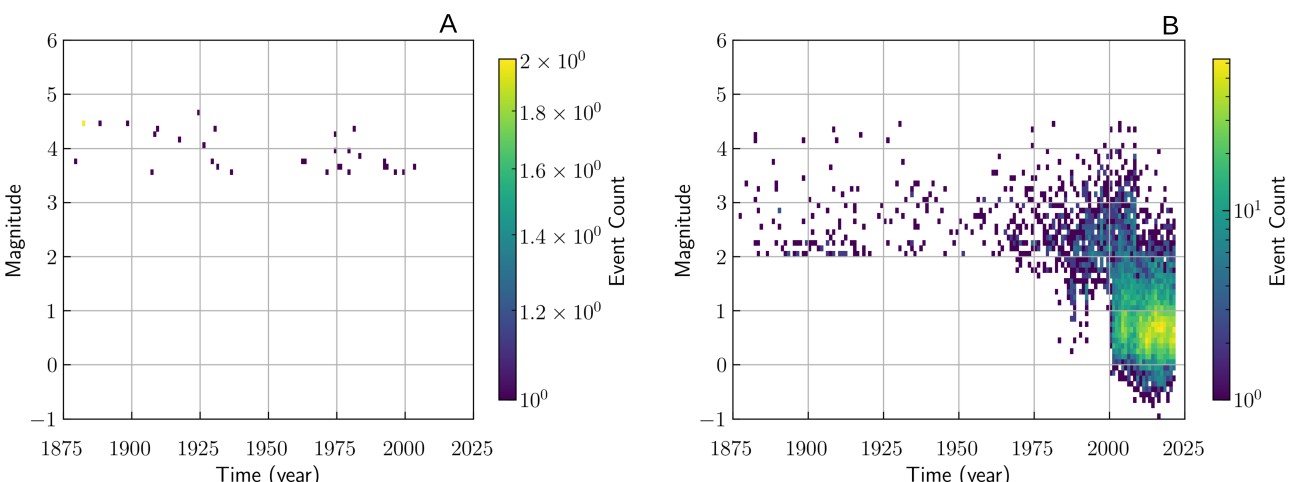

**Figure A3.** Magnitude-time density plots of the earthquakes in the (A) ESHM20 and (B) our catalogue north of 63 degrees latitude and between 15 and 25 degrees east longitude.
**IMT=PGA, kind=mean, Return period=475.0    IMT=PGA, kind=mean, Return period=2500.0**

**Figure A4.** ESHM20 seismic hazard maps for a return period of 475 years and 2500 years