# Peer review of "Probabilistic Seismic Hazard Assessment of Sweden"

_Natural Hazards and Earth System Sciences, 2023_

## Referee Comment (RC1)

Manuscript "Probabilistic seismic hazard assessment of Sweden"

by N. Joshi, B. Lund and R. Roberts

The manuscript presents a seismic hazard analysis for Sweden, a low-seismicity country in northern Europe. It includes information about seismicity, geology, the available earthquake catalog, and post-glacial faults there. The preprocessing of the earthquake catalog is explained in detail. The ground-motion logic tree is perused. As outcomes, the manuscript presents two seismic hazard maps for Sweden, with mean estimates for peak ground acceleration corresponding to return periods of 475 and 2500 years, and new hazard curves for four plus one sites. The previous seismic hazard analyses in the country are reviewed and the new results are compared to the new ones. The new hazard maps are also compared to the European Seismic Hazard Map 2020 (ESHM20). My main suggestion is to strengthen the results and align with the previous work for Sweden over 20 years ago by augmenting disaggregation. The line-by-line suggestions mainly deal with lesser issues.

Below I provide line-by-line suggestions:

Abstract

lines 10-11: "the high seismic activity on the post-glacial faults": this is meant in the national context, but since the calculated ground motion for the 475-yr return period barely reaches the threshold of engineering interest, 0.05g, it would be better to rephrase the expression, the same with "relatively high hazard"

1 Introduction

L37: in essence it is waste, the term "spent nuclear fuel" is also available

2 Earthquake activity in Sweden

L77: "Areas of high seismic activity", cf. above

L85-86: How do you know there are Burtraesk earthquakes among the pre-instrumental data?
-On L511 it is stated that macroseismic magnitudes have significant uncertainties. 4-4.5 does not appear that significant.

L90: The family name of the author is Muir Wood, not Wood.

3 Previous seismic hazard assessments for Sweden

L130-134: It seems that the most important previous hazard analysis for Sweden is by Wahlstroem and Gruenthal in the early 2000s. They provided disaggregation, which is the main argument for also providing disaggregation plots in the present work. The current version of the manuscript will be strengthened in the results part. Disaggregation is a basic calculation to identify the earthquake scenarios that contribute the most to a specified exceedance probability of ground-motion levels. It will add to the value of the work.

**4.2 Seismic source areas**

lines 152, 172, 208, 284, 285, 290 (possibly elsewhere as well):
Seismic source area (SSA) is not commonly used in PSHA. I would suggest replacing it with seismic source zone (SSZ) throughout the manuscript.

**4.4 Calculating recurrence parameters**

L345-346: "Although the first seismograph in Fennoscandia was installed in Sweden in 1904, the completeness magnitude of the catalogue has varied during the 20$^{th}$ century from about M4 to about M2." The first part and the second part of the sentence do not resonate well.

**4.6 Ground Motion Models**

L392 (report Goulet et al. 2018) the article for the NGA-East suite of GMPEs has been published:

Goulet CA, Bozorgnia Y, Kuehn N, Al Atik L, Youngs RR, Graves RW, Atkinson GM. NGA-East ground-motion characterization model part I: Summary of products and model development. Earthquake Spectra 37(S1), 1231-1282, 2021. https://doi.org/10.1177/87552930211018723

**5.1 Seismic hazard maps for Sweden**

L462: The input catalog spans 150 years according to Figure 3. How do the authors perceive the added value of the seismic hazard map with 2500-yr return period? Low exceedance probabilities imply rather high-magnitude earthquakes. Do you think the large earthquakes will occur in the areas with recurrent small earthquakes?

**5.2 Hazard curves for seismogenic areas**

L477-480: Four sites were picked up to represent areas of enhanced seismicity within the territory, and hazard curves are presented for the four sites in Figure 9.
Figure 1 shows five "sites of interest". What were the grounds for picking up the fifth site? It is located in an area with less seismicity than the other four sites.

**6 Discussion**

L498 "high seismic hazard", cf. above

Conclusions

L601-632: this is mostly more like a summary

Figures

Figure 1: add scale and/or coordinates to the map

The dashed black lines showing the Sorgenfrei-Tornquist etc. zones are very thin.

Caption: it is more conventional to separate the sites in the map and describe the symbols in the caption, what P, B, H, U, LV mean for instance.

Figure 2: The blue dots should be explained in the figure caption.

Figure 3: L223 states that your base catalog begins in 1375, supposedly this applies to Sweden as well. Did you remove dependent events and homogenize magnitude for data from 1875 onwards only? Better to repeat the years in the caption, now it is stated that this is all the Swedish earthquake data.

Figure 4: Coordinates are typically given on the figure frame. At least a scale should be provided.

Figure 6: Not all readers read the text from the beginning to the end, so writing more complete figure captions is an option to seriously consider throughout the manuscript.

Figure 7 caption states that "solid lines show the mean": cannot discern any solid lines in the figure

Figure 8: When displaying the two maps parallel, it is not ideal that the darkest shade refers to lower ground motion than the shade used for the largest ground-motion values.

There are four figures with the figure number preceded by the letter A (A1 to A4). Are they meant to constitute an Appendix? No reference to an appendix can be found.

The first three A-figures are referred to in the text, but Figure A4 is not.

Figure A2: a more logical scale would be gray – blue – green – orange – red showing the largest clusters

References

L837, L874, and others: BSSA abbreviation is misspelled: "Seimol"

L870: the author of this article is R. Muir Wood, not R. M. Wood, cf. L90

---

## Referee Comment (RC2)

**Review of the manuscript NHESS-2023-213**

The aim of the paper NHESS-2023-213 "Probabilistic Seismic Hazard Assessment of Sweden" is to present seismic hazard estimates (hazard maps and hazard curves for selected sites) for Sweden using the probabilistic seismic hazard analysis (PSHA). This country is characterised by low levels of seismicity and therefore the time length of the earthquake observations, which span a few hundreds of years in the best case, is much shorter than the seismic cycle of large earthquakes, which is of the order of thousands of years in low seismicity regions (e.g. Stein et al. 2015). Using sparse and limited sets of data represents a challenge to fully capture the epistemic uncertainties in a national seismic hazard model. In this context, the aim of this paper is of primary importance for seismic hazard analysis. However, there are some inaccuracies in the manuscript (e.g., the description of the steps for PSHA) and more explanations to justify the decisions taken by the authors to develop the seismic hazard model for Sweden are required. Furthermore, the English language seems to be quite poor in some paragraphs. Although I provide below some editorial comments on wording and sentences, I would suggest a significant revision in terms of the language throughout the manuscript.

Here I list the main technical and editorial points.

1- A discussion on the uncertainty in the parameters of the earthquake catalogue is not mentioned at all. What are the uncertainties in the epicentral locations and the magnitude? Are they accounted for in the estimation of the recurrence parameters?

2- The authors do not mention at all the focal mechanisms of the earthquakes in Sweden and Fennoscandia. Are there any focal mechanisms known for earthquakes that occurred in the region? Similarly, what is the hypocentral depth, together with the associated uncertainty, of the earthquakes in the final catalogue built for this work?

3- The discussion on the magnitude homogenisation and assessment of the completeness thresholds (Section 4.1.3) in the catalogue is difficult to follow and lacks crucial information. Is the $M_L$(HEL) used for all events in the final catalogue, including those from NORSAR, NNSN, and SNSN? If not, the description of how ML(HEL) was estimated is unnecessary. What are the equations used to convert ML into Mw? Are they applied to all the data in the final catalogue? For the assessment of the completeness threshold(s), from which year is the catalogue complete for Mc = 2 Mw? Furthermore, is a single Mc value used for the calculation of the recurrence parameters? Why did the authors not use the completeness thresholds for Fennoscandia estimated in ESHM20 or ESHM13?

4- The authors should explain better how they defined a Mmax distribution between 6.3 and 7.5 (I assume this is Mw, isn't it?). In analogue regions, there are no examples of 7.5 Mw earthquakes, so the authors should justify better the 7.5 Mw value.

5- If I have understood correctly, the authors have defined new TSZs and ASZs from the ESHM20. If this is the case, why did the authors use the TSZs and ASZs from ESHM20?

6- Is a single source model considered for the PSHA of Sweden? Alternative source models would account for different interpretations of the mapped tectonic structures, large-scale deformation, regional stress field, and observed seismicity in Sweden and Fennoscandia. It would ensure to capture the epistemic variability in the behaviour and location of seismogenic structures and their correlation with seismicity. Did the authors consider to use of the zoneless (zone-free, smoothed) models (see Beauval et al. 2006; Zechar and Jordan 2010 for more details) approach as an alternative sesmic model? This was included in the ESHM20 model and other national seismic hazard models, such as Germany (Grünthal et al. 2018) and France (Drouet et al. 2020).

7- How were the weights in the ground motion logic tree decided? Are there any available ground motion recordings for instrumental earthquakes in Sweden and Fennoscandia? If so, it would be useful to compare them with the predictions from the selected ground motion models. This comparison can be used to assign the weights for the ground motion models in the logic tree, together with expert judgements due to the limited ground motion dataset in the region.

8- Why was a minimum magnitude of 4.5 Mw selected for the hazard calculations? The minimum magnitude (Mmin) in a hazard calculation is defined as the threshold for potentially damaging earthquakes (e.g. Bommer and Crowley 2017). This parameter is usually defined between 4 and 5 Mw for PSHA. In the PSHA for the UK, it was set to 4.0 Mw because it includes the probability that the impulsive nature of small earthquakes and their high-frequency content could be potentially causing damage (Mosca et al., 2022). I would think that due to the low levels of seismicity in Sweden, this may be appropriate also for this country.

9- Section discussion (Section 6 here) should not repeat what was already written previously. It should emphasize the main result, highlight the strengths and limitations of the study, provide the interpretation of the results in the context of regional hazard and eventually give future research directions. For example, Subsection 6.2 "Comparison with previous studies" should be part of Section Results.

10- An acronym should be explained only when it is mentioned the first time in the manuscript. ML and Mw are not explained when they are used for the first time in Section 2.

11- All the geographical names mentioned in the text should be indicated on a map because not all the readers are familiar with the geography, geology and tectonics of Sweden.

Line 4: Include a comma before "which". Replace "large number of events" with " high number of events".

Line 5: Replace "5.9 to -1.4" with "-1.4 to 5.9". What is the magnitude scale?

Line 6: "less uncertainty" is in contradiction with the first line of the abstract, which states that the seismic hazard assessment in stable continental regions is challenging due to the limited amount of available data ". Also, "recurrence parameters to be calculated for more source areas than in previous studies" is unclear and I would suggest re-phrasing it.

Line 13: replace "highest PGAs" with "highest PGA values".

Line 19: What do the authors mean by "disaster development in the event"?

Line 25: Replace the full stop before Occurrence with a comma.

Line 26: Replace "as England and Jackson (2011) show, the risk" with "England and Jackson (2011) show that the risk".

Line 30: What is the magnitude scale in this case? How were these estimates (one event of magnitude 5 every 100 years and one event of magnitude 6 every 1000 years) computed? Replace "until 2005" with "before 2005".

Line 34: How large are the "large earthquakes"? Provide an indication.

Line 39: Provide references for "earlier estimates".

Line 44: Replace "The hazard is calculated using the OpenQuake engine (Pagani et al., 2014) and we produce hazard maps…" with "We use the OpenQuake engine (Pagani et al., 2014) to develop hazard maps …".

Line 49: The first sentence of Section 2 is more appropriate for the introduction than for this section, which could start with the second sentence. It would be useful to mention which are these damaging earthquakes and which damages were produced.

Line 76: For which year does the completeness magnitude of 0.5 correspond? What is the magnitude scale? In Section 2, both ML and Mw are used. Probably it is better to use only one magnitude scale, preferably Mw.

Line 82: How low is the magnitude? Provide an indication.

Figure 1: Besides reporting the geographical names in the text into Figure 1, it would be useful to show the distribution of the earthquakes in terms of magnitude highlighting those of magnitude 4 and above. Which magnitude scale is used in the figure, ML or Mw? Also, would it be possible to label the earthquakes mentioned in Section 2 into Figure 1, e.g. 1819 earthquake? Last, it is difficult to distinguish the Tornquist and Trans-European Suture zones from the earthquakes (they both are indicated by dots).

Section 3: In general, the main components (i.e. catalogue, source model, ground motion model) of each study, together with the highest hazard computed by the studies, should be explained to facilitate the comparison between models, including the model presented in the manuscript. Probably, a table which summarises the various components of previous studies in Sweden and Fennoscandia may be helpful. I recognise that indication of the resulting hazard in the previous studies is done, but not all the components are briefly described. For example, the ground motion models used in Bath (1979), Wahlström and Grünthal (2001), Mäntyniemi et al. (1993, 2001), etc are not indicated explicitly. The model of GSHAP and ESHM13 are cited but no information about them is provided. It would be useful to see how they differ from ESHM20 in terms of individual components and hazard results. Please indicate the magnitude scale every time (see lines 124-126).

Line 108: It would be useful to indicate how much "highest" is the highest hazard in Bath (1979).

Line 109: Replace "an ML ≥ 5 event" with "an event of 5 ML and above".

Lines 116: What does "various combinations of seismic source areas" mean? Also, replace "rate information" with "rate estimation".

Line 120: Move "for a probability of exceedance of $10^{-5}$ per year and a damping of 5%" at the end of the sentence. Furthermore, the damping is for spectral acceleration, not PGA.

Line 130: Replace "Wahlström and Grünthal (2000) and follow-up Wahlström and Grünthal (2001)" with "Wahlström and Grünthal (2000, 2001)".

Line 151: Provide the references for "two large PSHA projects for the nuclear industry in Finland".

Lines 152-153: Replace "The first, the Fennovoima project, assembled seismologists and geologists from Finland and Sweden to perform a full site-specific PSHA" with "In the Fennovoima project, seismologists and geologists from Finland and Sweden perform a full site-specific PSHA…".

Lines 169-170: Replace "events, from 1497 to 2014, with magnitudes 3.5 ≤ MW ≤ 5.8." with "events with magnitudes 3.5 ≤ MW ≤ 5.8 from 1497 to 2014.".

Lines 172-174: It is difficult to follow this sentence. I would suggest rephrasing it.

Line 173: Replace "In these zones," with "In ASZs with more than 30 earthquakes," and remove "for zones with more than 30 earthquakes" at the end of this sentence.

Line 177: It is double (not doubly) truncated Gutenberg-Richter. Correct it throughout the manuscript. Also, replace "using an automatic maximum" with "and an automatic maximum".

Line 179: Replace "is re-used" with "is assumed as a prior value".

Line 194: It should be mentioned that the GMM in Kotha et al. (2020) are for active shallow crustal regions in the ESHM20.

Section 4: The description of PSHA is inaccurate. It consists of four steps (e.g., Reiter, 1990; Baker et al., 2021): 1- Definition of seismic sources based on knowledge of the tectonics, geology and seismicity of the study area. 2- Quantification of the rate of earthquake occurrence for each seismic source zone using the Gutenberg-Richter frequency-magnitude law. 3- Characterise the 'earthquake effect' expressed in terms of some instrumental ground motion measure, such as PGA, or seismic intensity. 4- Estimation of the hazard at the site(s) by analytically integrating over the source models for the location and size of potential future earthquakes (Steps 1 and 2) with expected values of the potential shaking intensity caused by these future earthquakes (Step 3), including the associated variability in each. The development of the earthquake catalogue is part of step 1.

Subsections 4.1 and 4.2: They can be merged. Why few events from the ESHM20 catalogue are not included in the FENCAT catalogue? When did these events occur? and what was their magnitude? How small were the events in the SNSN catalogue that were included in the final catalogue? How were quarry, industrial or military blasts, rock bursts, mine collapses etc identified as nontectonic earthquakes? Did the authors remove also non-tectonic events offshore?

Line 231: Indicate the magnitude range for the 24,215 events in the final catalogue.

Section 4.1.2: How do the results of the declustering method (modified Gardner and Knopoff, 1974) compare with that from the method of Burkhard and Grünthal (2009) that was calibrated for the earthquake catalogue in Central Europe and was used in ESHM13 and ESHM20?

Line 223: Replace "at our disposal spans the year 1375 until the end of" with "that we used spans between 1375 and the end of".

Line 234: Provide a reference for the first sentence.

Lines 249-251: It is difficult to follow this sentence, so I would suggest rephrasing it.

Line 257: Replace "a smaller fraction of dependent events of only 11%, a difference to our result which is likely due to the fact that" with "less dependent events than those in our study. The difference (11%) is probably because".

Line 249: How do the earthquakes in the FENCAT compare with those in the ESHM20 catalogue in terms of epicentral location and magnitude? In Figure 4 the earthquakes should be plotted in terms of magnitude to facilitate this comparison.

Figure 2: I would suggest adding an extra figure to show the distribution of the seismic source model. Figure 2 should show only the final catalogue for this work where the distribution of earthquakes in terms of magnitude should be highlighted.

Line 312: Replace "is complicated by" with "is difficult due to".

Line 315: Delete "purposes".

Lines 320 and the following lines: Indicate the magnitude scale.

Table 2: Why aren't the recurrence parameters of all ASZs reported in this table? The *a* and b-values for zones 1,4-8,10-12, etc are missing. For transparency, they all should be reported. Is the activity rate computed for 0 Mw? It would be also useful to indicate how many earthquakes within the completeness thresholds were used to estimate the recurrence parameters. As mentioned before, it is unclear which completeness thresholds were used for the estimation of the recurrence parameters. For many zones, the b-value seems to be quite low (< 0.9), what is the reason for this? How do the b-values compare with previous studies, in particular the ESHM20 for similar zones?

Line 383: replace "construct" with "develop".

Line 415: What is Model 5?

Line 433: Replace "yearly" with "annual".

Lines 441-444: The hypocentral depths of the earthquakes in the catalogue have not been discussed at all in the manuscript to justify the depth distribution indicated here for the hazard calculations.

Section 4.7: Openquake requires also the definition of the faulting style for potential, future earthquakes, defined by rake, dip and strike. This set of parameters has not been discussed at all in the manuscript.

Line 456: In the revision of the Eurocode 8, the seismic hazard is described in terms of the 5% damped maximum spectral acceleration at a short period and 1.0 s period, and PGA is not mentioned anymore. Would it be useful to estimate national maps also for spectral acceleration for a representative short period (e.g. 0.2 s) and 1.0 s?

Figure 10: What is plotted in Figure 10 exactly? Is this the relative or absolute difference between the new map for Sweden and the ESHM20 maps? It would be helpful to produce such a map also for ESHM20 and the other previous maps discussed in Section 6.2.

**References**

BAKER, JW, BRADLEY, BA, AND STAFFORD, PJ. 2021. *Seismic Hazard and Risk Analysis.* (Cambridge University Press).

BEAUVAL C, SCOTTI O, BONILLA F. 2006. THE ROLE OF SEISMICITY MODELS IN PROBABILISTIC SEISMIC HAZARD ESTIMATION: COMPARISON OF A ZONING AND A SMOOTHING APPROACH, *GEOPHYSICAL JOURNAL INTERNATIONAL*, VOL. 165(2):584–595.

BOMMER, JJ, and CROWLEY, H. 2017. The purpose and definition of the minimum magnitude limit in PSHA calculations, *Seismological Research Letters,* Vol. 88, 1097-1106.

DROUET, S, AMERI, G, LE DORTZ, K, SECANELL, R, and SENFAUTE, G. 2020. A probabilistic seismic hazard map for the metropolitan France, *Bulletin of Earthquake Engineering*, doi.org/10.1007/s10518-020-900790-7.

GRÜNTHAL, G, STROMEYER, D, BOSSE, C, COTTON, F, and BINDI, D. 2018. The probabilistic seismic hazard assessment of Germany—version 2016, considering the range of epistemic uncertainties and aleatory variability, *Bulletin of Earthquake Engineering,* Vol. 16(10)**,** 4339-4395.

MOSCA, I, SARGEANT, S, BAPTIE, B, MUSSON, RMW, and PHARAOH, T. 2022. The 2020 national seismic hazard model for the United Kingdom, *Bulletin of Earthquake Engineering*, Vol. 20, 633–675.

REITER, L. 1990. Earthquake Hazard Analysis (New York: Columbia University Press).

STEIN S, LIU M, CAMELBEECK T, MERINO M, LANDGRAF A, HINTERSBERGER E, KUEBLER S. 2015. Challenges in assessing seismic hazard in intraplate Europe, In *Seismicity, Fault rupture and earthquake hazards in slowly deforming region*, LANDGRAF A, KUEBLER S, HINTERSBERGER E, STEIN S (Eds), Geological Society Special Publication, Vol. 432, 29‑32.

ZECHAR JD, and JORDAN TH. 2010. Simple smoothed seismicity earthquake forecasts for Italy, *Annali di Geofisica*, Vol. 53(3), 99–105

---

## Author Comment (AC1)

We thank Anonymous Referee #1 for going through the manuscript and providing feedback on what is insufficient and can be improved. Below, the reviewer comments and line-by-line suggestions are in black. Our responses to their comments are in blue.

The manuscript presents a seismic hazard analysis for Sweden, a low-seismicity country in northern Europe. It includes information about seismicity, geology, the available earthquake catalog, and post—glacial faults there. The preprocessing of the earthquake catalog is explained in detail. The ground—motion logic tree is perused. As outcomes, the manuscript presents two seismic hazard maps for Sweden, with mean estimates for peak ground acceleration corresponding to return periods of 475 and 2500 years, and new hazard curves for four plus one sites. The previous seismic hazard analyses in the country are reviewed and the new results are compared to the new ones. The new hazard maps are also compared to the European Seismic Hazard Map 2020 (ESHM20). My main suggestion is to strengthen the results and align with the previous work for Sweden over 20 years ago by augmenting disaggregation. The line-by-line suggestions mainly deal with lesser issues.

We add disaggregation plots and respond in more detail to this issue below.

Abstract lines 10-11: "the high seismic activity on the post—glacial faults": this is meant in the national context, but since the calculated ground motion for the 475-yr return period barely reaches the threshold of engineering interest, 0.05g, it would be better to rephrase the expression, the same with "relatively high hazard"

We agree that the seismic activity is not high in a global context. We have now rephrased the sentence as "This is in contrast to previous studies, which have not considered the relatively high seismic activity on the post-glacial faults. We also find the hazard to be relatively high along the northeast coast and in southwestern Sweden, whereas the southeast and the mountain region to the northwest have a relatively low hazard."

1 Introduction
L37: in essence it is waste, the term "spent nuclear fuel" is also available
We take the suggestion on board and have modified the manuscript as follows : "For sensitive infrastructure sites such as nuclear power plants, repositories for spent nuclear-fuel, dams and mines, seismic hazard estimates are required and there is therefore a need to better define the hazard spatially, estimate potential ground motions and investigate associated uncertainties."

2 Earthquake activity in Sweden
L77: "Areas of high seismic activity", cf. above
We have rephrased the sentence as follows: "Areas of relatively high seismic activity include the southwestern part of Sweden across Lake Vänern, along the northeast coast and in the far north, see Figure 1."

L85-86: How do you know there are Burtraesk earthquakes among the pre—instrumental data?
We know this from the FENCAT catalogue (Ahjos and Uski, 1992), which is a catalogue of earthquakes in northern Europe compiled from all the available historical publications,

catalogues, studies and reports from the region. The epicentral coordinates of historical earthquakes (1375-1964) were either reported by the sources from which the data was obtained or were estimated from geographic maps and observation reports. The epicentre for macroseismic events was recorded as the centre of the area of perceptibility or at the site with the strongest intensity observed. Epicentral coordinates were derived from isoseismal maps in the case of detailed macroseismic studies.

-On L511 it is stated that macroseismic magnitudes have significant uncertainties. 4-4.5 does not appear that significant.

We have not studied the historical Burträsk events ourselves, but there was a ML 3.9 event in 1907 and a ML 4.4 event in 1909, where the macroseismic data probably is reasonably good. For earlier events in the Fencat the uncertainties are considered larger.

L90: The family name of the author is Muir Wood, not Wood.
The citation has been updated and appears in the text as follows : "a MS 5.6 event occurred in the waters between Sweden and Denmark (Muir Wood, 1989)"

3 Previous seismic hazard assessments for Sweden
L130—134: It seems that the most important previous hazard analysis for Sweden is by Wahlstroem and Gruenthal in the early 2005. They provided disaggregation, which is the main argument for also providing disaggregation plots in the present work. The current version of the manuscript will be strengthened in the results part. Disaggregation is a basic calculation to identify the earthquake scenarios that contribute the most to a specified exceedance probability of ground—motion levels. It will add to the value of the work.

A new figure has been added to the manuscript, similar to that provided by Wahlström and Gruenthal in their study. This figure plots the results of the disaggregation for two sites, one being the site at which W&G performed their disaggregation and the other being the site at which we estimate the hazard to be the highest. The text has been modified to reflect what is seen in the figure.

4.2 Seismic source areas
lines 152, 172, 208, 284, 285, 290 (possibly elsewhere as well): Seismic source area (SSA) is not commonly used in PSHA. I would suggest replacing it with seismic source zone (SSZ) throughout the manuscript.
The terminology has been updated throughout the manuscript.

4.4 Calculating recurrence parameters
L345—346: "Although the first seismograph in Fennoscandia was installed in Sweden in 1904, the completeness magnitude of the catalogue has varied during the 20th century from about M4 to about M2." The first part and the second part of the sentence do not resonate well.
We agree and the following changes have been made to the manuscript: "Estimating recurrence parameters is challenging in a low seismicity area like Fennoscandia, where larger earthquakes are rare and population density low. Very few events have been recorded prior to the installation of more sensitive seismic networks since the 1960s and 1970s. The

completeness magnitude of the national earthquake-catalogue has varied during the 20th century between about M4 to about M2."

4.6 Ground Motion Models
L392 (report Goulet et al. 2018) the article for the NGA—East suite of GMPEs has been published: Goulet CA, Bozorgnia Y, Kuehn N, AI Atik L, Youngs RR, Graves RW, Atkinson GM. NGA-East ground- motion characterization model part I: Summary of products and model development. Earthquake Spectra 37(51), 1231—1282, 2021. https://doi.org/10.1177/87552930211018723
The reference has been updated.

5.1 Seismic hazard maps for Sweden
L462: The input catalog spans 150 years according to Figure 3. How do the authors perceive the added value of the seismic hazard map with 2500-yr return period? Low exceedance probabilities imply rather high—magnitude earthquakes. Do you think the large earthquakes will occur in the areas with recurrent small earthquakes?

The 475-year and 2500-year hazard map were produced as they're the most commonly used. As with most global catalogues around the world with seismic instrumental records dating back to early 20th century, it is tricky to truly estimate the potential of the rarer less-frequent earthquakes. However, in our case, ruptures of magnitude 7 as recently as 700 years ago have been estimated to have occurred on the Stuoragurra fault in northern Norway by Olesen et al., 2021. We therefore consider it reasonable to include the 2500-year RP seismic hazard map.
It is difficult to comment with certainty on whether large earthquakes would occur in areas with recurrent small earthquakes, an issue that is common to all stable continental regions. We elaborate on this in the discussion section 6.3 of the manuscript.

5.2 Hazard curves for seismogenic areas
L477-480: Four sites were picked up to represent areas of enhanced seismicity within the territory, and hazard curves are presented for the four sites in Figure 9. Figure 1 shows five "sites of interest". What were the grounds for picking up the fifth site? It is located in an area with less seismicity than the other four sites.
The fifth site is Uppsala, which is part of a region where no relatively significant difference is seen between the ESHM20 results and our hazard estimates. This allows us to study the spread of the hazard curves, shown in Figure 7, and choose the right weights for the ground-motion model logic tree.
Combining feedback from below, Figure 1 is now modified and accounts for the issues raised by the reviewer here.

6 Discussion
L498 "high seismic hazard", cf. above
Modified as per suggestions by the reviewer to "The inference that the last 20 years of earthquake data in northern Sweden indicates that the area has relatively high seismic hazard is interesting in the light of recent work on the PGFs in the region."

Conclusions
L601-632: this is mostly more like a summary

We have now renamed the conclusions section to summary, to accurately reflect what lies therein.

Figures
Figure 1: add scale and/or coordinates to the map
Modified as per reviewer's suggestion.

The dashed black lines showing the Sorgenfrei—Tornquist etc. zones are very thin.
Modified as per reviewer's suggestion.
Caption: it is more conventional to separate the sites in the map and describe the symbols in the caption, what P, B, H, U, LV mean for instance.
Modified as per reviewer's suggestion, as described further above.

Figure 2: The blue dots should be explained in the figure caption.
Modified as per reviewer's suggestion. The caption now says "Tectonic source zonation scheme (A) and the area source zonation scheme (B) used in this study. The red line indicates a zone encompassing Sweden that is 300 km from the Swedish border or economic zone boundaries. Blue dots show the earthquakes. Numbers detail the zone numbers.".

Figure 3: L223 states that your base catalog begins in 1375, supposedly this applies to Sweden as well. Did you remove dependent events and homogenize magnitude for data from 1875 onwards only? Better to repeat the years in the caption, now it is stated that this is all the Swedish earthquake data.
The caption is now modified to account for the reviewer's concerns and says the following: "Magnitude-time density plot of the earthquakes recorded to have occurred in the Swedish economic zone since 1875 (Blue dots within the red zone in Figure \ref{fig:zones}) from the declustered and magnitude homogenized catalogue."

Figure 4: Coordinates are typically given on the figure frame. At least a scale should be provided.
Modified as per reviewer's suggestion.

Figure 6: Not all readers read the text from the beginning to the end, so writing more complete figure captions is an option to seriously consider throughout the manuscript.
Modified as per reviewer's suggestion.
We consider that the captions describe what is shown in the figures.

Figure 7 caption states that "solid lines show the mean": cannot discern any solid lines in the figure
This was erroneously stated previously and has now been modified as per reviewer's suggestion to say the following:"Mean hazard curves for Uppsala, Sweden, for different GMM implementations, see Section 4.6 for details. Yearly probability of exceedance versus PGA in g. Solid symbols show the mean, while the upper and lower dashed lines show the 0.84 and 0.16 fractiles."

Figure 8: When displaying the two maps parallel, it is not ideal that the darkest shade refers to lower ground motion than the shade used for the largest ground-motion values.

Using the same scale range for 475-year and 2500-year hazard maps makes interpreting the 475-year hazard map almost impossible. We therefore choose to continue using different scale bars for the two hazard maps instead,  but have adjusted the range for the 2500-year map.

There are four figures with the figure number preceded by the letter A (A1 to A4). Are they meant to constitute an Appendix? No reference to an appendix can be found.
The first three A—figures are referred to in the text, but Figure A4 is not.
This is probably a formatting issue with LaTeX and the authors have added an appendix section heading to alleviate the reviewer's concerns.

Figure A2: a more logical scale would be gray — blue - green - orange - red showing the largest clusters
Modified as per reviewer's suggestion.

References
L837, L874, and others: BSSA abbreviation is misspelled: "Seimol"
Modified as per reviewer's suggestion.
L870: the author of this article is R. Muir Wood, not R. M. Wood, cf. L90
Modified as per reviewer's suggestion.

---

## Author Comment (AC2)

We thank Dr. Ilaria Mosca for her review and comments. Below are Dr. Mosca's comments and line-by-line suggestions, in black. Our responses are in blue.

The aim of the paper NHESS-2023-213 "Probabilistic Seismic Hazard Assessment of Sweden" is to present seismic hazard estimates (hazard maps and hazard curves for selected sites) for Sweden using the probabilistic seismic hazard analysis (PSHA). This country is characterised by low levels of seismicity and therefore the time length of the earthquake observations, which span a few hundreds of years in the best case, is much shorter than the seismic cycle of large earthquakes, which is of the order of thousands of years in low seismicity regions (e.g. Stein et al. 2015). Using sparse and limited sets of data represents a challenge to fully capture the epistemic uncertainties in a national seismic hazard model. In this context, the aim of this paper is of primary importance for seismic hazard analysis. However, there are some inaccuracies in the manuscript (e.g., the description of the steps for PSHA) and more explanations to justify the decisions taken by the authors to develop the seismic hazard model for Sweden are required. Furthermore, the English language seems to be quite poor in some paragraphs. Although I provide below some editorial comments on wording and sentences, I would suggest a significant revision in terms of the language throughout the manuscript. Here I list the main technical and editorial points.

1- A discussion on the uncertainty in the parameters of the earthquake catalogue is not mentioned at all. What are the uncertainties in the epicentral locations and the magnitude? Are they accounted for in the estimation of the recurrence parameters?
Location and magnitude uncertainties have of course evolved over time, they are currently 2 -3 km and 0.1 – 0.2 on average in the Swedish seismic network. The historical, macroseismic data has significantly larger uncertainties. We do not consider the location accuracies to be a problem, concerning the size of the seismic source areas. The magnitude uncertainties for events in the older data does affect the recurrence calculations but are accounted for in the logic tree through the a- and b-value 2-sigma uncertainties. We have now added information on uncertainties to line 221 (in the original manuscript).

2- The authors do not mention at all the focal mechanisms of the earthquakes in Sweden and Fennoscandia. Are there any focal mechanisms known for earthquakes that occurred in the region? Similarly, what is the hypocentral depth, together with the associated uncertainty, of the earthquakes in the final catalogue built for this work?
We do refer to Gregersen et al. (2021) for a recent review of the debate surrounding the source of Fennoscandian seismicity, which includes focal mechanism studies and their interpretations. As for the depth distribution of earthquakes, Gregersen et al. (2021) says the following "Information on the focal depth distribution is not optimal due to the combination of sparse station density and large lateral variations in the crustal structure. Routine source depth estimates may contain significant uncertainties, and fixed depth estimates are frequently used by some seismic observatories." Earthquakes in Sweden are mostly strike-slip, and focal depths vary widely between near-surface and down to 35 km depth. We added more info on depth after line 87, and also refer to the depth implementation in OpenQuake discussed at line 441, and focal mechanisms info after line 95 (original manuscript).

3- The discussion on the magnitude homogenisation and assessment of the completeness thresholds (Section 4.1.3) in the catalogue is difficult to follow and lacks crucial information. Is the ML(HEL) used for all events in the final catalogue, including those from NORSAR, NNSN, and SNSN? If not, the description of how ML(HEL) was estimated is unnecessary. What are the equations used to convert ML into Mw? Are they applied to all the data in the final catalogue? For the assessment of the completeness threshold(s), from which year is the catalogue complete for Mc = 2 Mw? Furthermore, is a single Mc value used for the calculation of the recurrence parameters? Why did the authors not use the completeness thresholds for Fennoscandia estimated in ESHM20 or ESHM13?
There was an unfortunate error in the equations in line 270, as they should say Mw(HEL) =…
We have corrected this and added more text to the section, making it clear that all magnitudes are converted to Mw(HEL), which is what we later refer to as just Mw. Completeness vary in time and per zone, we have made individual assessments for different source zones and time periods, as noted on line 349. For the entire data set, completeness is around Mw 2 for onshore areas from around the mid-1970s. We did not use the completeness thresholds from ESHM20 as they have a magnitude cut-off of Mw 3.5 and therefore do not include many Swedish events.

4- The authors should explain better how they defined a Mmax distribution between 6.3 and 7.5 (I assume this is Mw, isn't it?). In analogue regions, there are no examples of 7.5 Mw earthquakes, so the authors should justify better the 7.5 Mw value.
As we write in section 4.3, the rupture of post-glacial faults in Northern-Sweden has been estimated to lead to earthquakes as large as Mw8 nearly 10,000 years ago and the most recent rupture of the post-glacial faults in Northern-Norway was estimated to be at M7.0. We therefore consider it reasonable to include Mmax=7.5 with a low weight of 0.05 on the logic tree.

5- If I have understood correctly, the authors have defined new TSZs and ASZs from the ESHM20. If this is the case, why did the authors use the TSZs and ASZs from ESHM20?
Yes, we do define new source zones compared to ESHM20 but no, we do not use the ESHM20 zones, we use the newly defined zones. This is described in section 4.2,

6- Is a single source model considered for the PSHA of Sweden? Alternative source models would account for different interpretations of the mapped tectonic structures, large-scale deformation, regional stress field, and observed seismicity in Sweden and Fennoscandia. It would ensure to capture the epistemic variability in the behaviour and location of seismogenic structures and their correlation with seismicity. Did the authors consider to use of the zoneless (zone-free, smoothed) models (see Beauval et al. 2006; Zechar and Jordan 2010 for more details) approach as an alternative sesmic model? This was included in the ESHM20 model and other national seismic hazard models, such as Germany (Grünthal et al. 2018) and France (Drouet et al. 2020).
Yes, we only use one source model. We have an ongoing project where we use seismicity smoothing as implemented by Frankel 1995, to estimate seismic hazard. However, we decided to not include that in this study owing to time-constraints for a PhD-student. We aim to include smoothed seismicity in the future as a part of a project on a joint Fennoscandian seismic hazard assessment.

7- How were the weights in the ground motion logic tree decided? Are there any available ground motion recordings for instrumental earthquakes in Sweden and Fennoscandia? If so, it would be useful to compare them with the predictions from the selected ground motion models. This comparison can be used to assign the weights for the ground motion models in the logic tree, together with expert judgements due to the limited ground motion dataset in the region.

The ground motion logic tree includes the ground motion models included in ESHM20 and the regional ground motion model FennoG16, developed by our colleagues in Finland. The combined weights of the ESHM20 GMMs were scaled down to include the FennoG16. The actual weights were then decided based on spread of the uncertainties in the ground motion curves for Uppsala, as shown in figure 7, where we aim for weights that give us fractile curves wider than the FennoG16 but narrower than the ESHM20 logic tree.

FennoG16 (Fülöp et al., 2020) used all available ground motion data in Fennoscandia for its development. There are no strong motion data from Fennoscandia, and very little data from events larger than M4, or close to events larger than M3.5. FennoG16 therefore also includes data from Eastern North America, as does the ESHM20 craton GMM (Weatherill & Cotton, 2020) who also compare the ESHM20 craton model to FennoG16. This is noted in section 4.6.

8- Why was a minimum magnitude of 4.5 Mw selected for the hazard calculations? The minimum magnitude (Mmin) in a hazard calculation is defined as the threshold for potentially damaging earthquakes (e.g. Bommer and Crowley 2017). This parameter is usually defined between 4 and 5 Mw for PSHA. In the PSHA for the UK, it was set to 4.0 Mw because it includes the probability that the impulsive nature of small earthquakes and their high-frequency content could be potentially causing damage (Mosca et al., 2022). I would think that due to the low levels of seismicity in Sweden, this may be appropriate also for this country.

We noted that it is common to use a minimum magnitude of 4 Mw when it comes to a seismic hazard assessment focussed on the civil infrastructure, while a minimum magnitude of 5 is commonly used when estimating seismic hazard for nuclear power installations. We therefore believed a minimum magnitude of Mw 4.5 was a reasonable compromise between the two cases. However, we have now calculated the seismic hazard for a minimum magnitude of Mw 4.0 and present the results below, as well as include them in the manuscript.

[Figure]

Seismic hazard map with a minimum magnitude of Mw 4 for a return period of 475 years.

[Figure]

Seismic hazard map with a minimum magnitude of Mw 4 for a return period of 2500 years.

9- Section discussion (Section 6 here) should not repeat what was already written previously. It should emphasize the main result, highlight the strengths and limitations of the study, provide the interpretation of the results in the context of regional hazard and eventually give future research directions. For example, Subsection 6.2 "Comparison with previous studies" should be part of Section Results.

We emphasise the main results in the Results section. Our intention with the discussion section was to contextualise the results in terms of the challenges we face to estimate recurrence parameters, other studies that have estimated the hazard in Sweden, and other studies that comment on the regional stress field and tectonics. We do agree that there is some degree of repetition in this section and modifications have been made to address this. We do not agree that Subsection 6.2 "Comparison with previous studies" should be part of Section Results as we want the results section to highlight our findings.

10- An acronym should be explained only when it is mentioned the first time in the manuscript. ML and Mw are not explained when they are used for the first time in Section 2.
Changes have been made to address this concern.

11- All the geographical names mentioned in the text should be indicated on a map because not all the readers are familiar with the geography, geology and tectonics of Sweden.
The text and figures have been appropriately modified.

Line 4: Include a comma before "which". Replace "large number of events" with " high number of events".
Modified as per reviewer's suggestion.
Line 5: Replace "5.9 to -1.4" with "-1.4 to 5.9". What is the magnitude scale?
Modified as per reviewer's suggestion.
Line 6: "less uncertainty" is in contradiction with the first line of the abstract, which states that the seismic hazard assessment in stable continental regions is challenging due to the limited amount of available data ". Also, "recurrence parameters to be calculated for more source areas than in previous studies" is unclear and I would suggest re-phrasing it.
 "less uncertainty" refers to a comparison to earlier studies and not to an overall challenge. We keep that and rephrase the "more source areas" part
Line 13: replace "highest PGAs" with "highest PGA values".
Modified as per reviewer's suggestion.
Line 19: What do the authors mean by "disaster development in the event"?
Modified and re-phrased.
Line 25: Replace the full stop before Occurrence with a comma.
Modified as per reviewer's suggestion.
 Line 26: Replace "as England and Jackson (2011) show, the risk" with "England and Jackson (2011) show that the risk".
Modified as per reviewer's suggestion.
Line 30: What is the magnitude scale in this case? How were these estimates (one event of magnitude 5 every 100 years and one event of magnitude 6 every 1000 years) computed? Replace "until 2005" with "before 2005".
Added Mw, we refer to the reference for details of computations.
Line 34: How large are the "large earthquakes"? Provide an indication.
Modified as per reviewer's suggestion.
Line 39: Provide references for "earlier estimates".
As this is outlining what is to come in the paper we refrain from including the 16 references here. They are discussed in section 3.
Line 44: Replace "The hazard is calculated using the OpenQuake engine (Pagani et al., 2014) and we produce hazard maps…" with "We use the OpenQuake engine (Pagani et al., 2014) to develop hazard maps …".
Modified as per reviewer's suggestion.

Line 49: The first sentence of Section 2 is more appropriate for the introduction than for this section, which could start with the second sentence. It would be useful to mention which are these damaging earthquakes and which damages were produced.

We think the first sentence is appropriate in a section on Earthquake activity and keep it here. The damaging (large) earthquakes are discussed further down in the section, we added damage descriptions there.

Line 76: For which year does the completeness magnitude of 0.5 correspond? What is the magnitude scale? In Section 2, both ML and Mw are used. Probably it is better to use only one magnitude scale, preferably Mw.
Mc of 0.5 refers to the modified network after 2000, we added that and ML. As not all magnitudes in the section are available as Mw or ML, we use both but made it clear.

Line 82: How low is the magnitude? Provide an indication.
Modified as per reviewer's suggestion.

Figure 1: Besides reporting the geographical names in the text into Figure 1, it would be useful to show the distribution of the earthquakes in terms of magnitude highlighting those of magnitude 4 and above. Which magnitude scale is used in the figure, ML or Mw? Also, would it be possible to label the earthquakes mentioned in Section 2 into Figure 1, e.g. 1819 earthquake? Last, it is difficult to distinguish the Tornquist and Trans-European Suture zones from the earthquakes (they both are indicated by dots).
The figure uses the homogenized moment magnitude scale and we include the large earthquakes mentioned in Section2. Highlighting all events above M4 would clutter the Figure too much, instead we show those in Figure 2A. The Tornquist and Trans-European Suture Zones are better outlined.

Section 3: In general, the main components (i.e. catalogue, source model, ground motion model) of each study, together with the highest hazard computed by the studies, should be explained to facilitate the comparison between models, including the model presented in the manuscript. Probably, a table which summarises the various components of previous studies in Sweden and Fennoscandia may be helpful. I recognise that indication of the resulting hazard in the previous studies is done, but not all the components are briefly described. For example, the ground motion models used in Bath (1979), Wahlström and Grünthal (2001), Mäntyniemi et al. (1993, 2001), etc are not indicated explicitly. The model of GSHAP and ESHM13 are cited but no information about them is provided. It would be useful to see how they differ from ESHM20 in terms of individual components and hazard results. Please indicate the magnitude scale every time (see lines 124-126).
We choose to highlight our study and results first, followed by a comparison with what was obtained by ESHM20. We compare the methodological differences through out the paper with the intention of making it easier for the reader to follow the thread when it comes to understanding why our results are similar and different. We appreciate the reviewers suggestion but choose to only provide results from the previous studies and not further discuss the other components or choices made.

Line 108: It would be useful to indicate how much "highest" is the highest hazard in Bath (1979).
This has now been rephrased to indicate that Båth calculated earthquake risk according to the formula put forward by Lomnitz around 1950s-60s describing earthquake risk as"*the probability R(D/T) that a shock of mean return period T occurs during a design period D.*" There are broader similarities between Båth's estimates of seismic risk and our calculations.

Line 109: Replace "an ML ≥ 5 event" with "an event of 5 ML and above".
Modified as per reviewer's suggestion.

Lines 116: What does "various combinations of seismic source areas" mean? Also, replace "rate information" with "rate estimation".
*Modified as per reviewer's suggestion. The text reads as follows:*
*FENCAT data until 1987 were used for the first Swedish probabilistic seismic hazard assessment (PSHA) work, directed at site-specific assessments for Sweden's four nuclear power plants (SKI, 1992). Fennoscandian earthquakes south of latitude 61◦ were used to estimate the seismicity rate.*

Line 120: Move "for a probability of exceedance of 10−5 per year and a damping of 5%" at the end of the sentence. Furthermore, the damping is for spectral acceleration, not PGA.
Modified as per reviewer's suggestion. PGA in SKI92:3 is defined at 100 Hz and used also for spectral acceleration, which is why we used it here.

Line 130: Replace "Wahlström and Grünthal (2000) and follow-up Wahlström and Grünthal (2001)" with "Wahlström and Grünthal (2000, 2001)".
Modified as per reviewer's suggestion.
Line 151: Provide the references for "two large PSHA projects for the nuclear industry in Finland".
The two are referenced in the next sentences. No change.

Lines 152-153: Replace "The first, the Fennovoima project, assembled seismologists and geologists from Finland and Sweden to perform a full site-specific PSHA" with "In the Fennovoima project, seismologists and geologists from Finland and Sweden perform a full site-specific PSHA…".
No change, as the sentence follows on the previous.

Lines 169-170: Replace "events, from 1497 to 2014, with magnitudes 3.5 ≤ MW ≤ 5.8." with "events with magnitudes 3.5 ≤ MW ≤ 5.8 from 1497 to 2014.".
Stylistic, no change.

Lines 172-174: It is difficult to follow this sentence. I would suggest rephrasing it. '
The sentence is rephrased as follows

*We split the sentence in two: Fennoscandia is assigned to a single maximum magnitude zone,*
*within which Mmax is uniform. It is divided into two "completeness zones" (CSZ) where reporting is assumed to be homogeneous through time such that the temporal variation in the magnitude of completeness, Mc , is the same all through the zone.*

Line 173: Replace "In these zones," with "In ASZs with more than 30 earthquakes," and remove "for zones with more than 30 earthquakes" at the end of this sentence.

Sentences modified

Line 177: It is double (not doubly) truncated Gutenberg-Richter. Correct it throughout the manuscript. Also, replace "using an automatic maximum" with "and an automatic maximum".
There seems to be different opinions about "doubly/double". Early investigators, e.g. Cosentino et al. (1977) and Kijko & Sellevoll (1998) uses doubly, later investigators vary their use and Danciu et al. (2021) uses both. We stay with doubly. No change.

Line 179: Replace "is re-used" with "is assumed as a prior value".
No change, as the TSZ b-values are used as-is for the ASZs with few events, not as a prior for further calculations.

Line 194: It should be mentioned that the GMM in Kotha et al. (2020) are for active shallow crustal regions in the ESHM20.
Modified as per reviewer's suggestion.

Section 4: The description of PSHA is inaccurate. It consists of four steps (e.g., Reiter, 1990; Baker et al., 2021): 1- Definition of seismic sources based on knowledge of the tectonics, geology and seismicity of the study area. 2- Quantification of the rate of earthquake occurrence for each seismic source zone using the Gutenberg-Richter frequency-magnitude law. 3- Characterise the 'earthquake effect' expressed in terms of some instrumental ground motion measure, such as PGA, or seismic intensity. 4- Estimation of the hazard at the site(s) by analytically integrating over the source models for the location and size of potential future earthquakes (Steps 1 and 2) with expected values of the potential shaking intensity caused by these future earthquakes (Step 3), including the associated variability in each. The development of the earthquake catalogue is part of step 1.
There are a number of ways to divide up the PSHA methodology, Baker himself used 5 points in 2013. Our 3 points incorporate the important points, but we realize that the text is a bit too brief, considering we spend 7 subsections describing our work. The text has been modified to better reflect the subsections.

Subsections 4.1 and 4.2: They can be merged. Why few events from the ESHM20 catalogue are not included in the FENCAT catalogue? When did these events occur? and what was their magnitude? How small were the events in the SNSN catalogue that were included in the final catalogue? How were quarry, industrial or military blasts, rock bursts, mine collapses etc identified as nontectonic earthquakes? Did the authors remove also non-tectonic events offshore?
We think the reading is easier if the subsections are kept specific so as to not get too long. The ESHM20 catalogue was prepared by aggregating earthquakes from several catalogues across Europe whereas the FENCAT catalogue includes data specifically from Fennoscandia, which is a reason why not all events that exist in the catalogue used by ESHM20 exist in the FENCAT catalogue. We, therefore, include those events from ESHM20 that lie within our source zones and were missing from FENCAT. This includes a M3.81 earthquake that occurred in northern Germany in 1909, two earthquakes with magnitudes 4.57 and 4.73 from 2004 that occurred in Kaliningrad, four events in Latvia with magnitudes between 4 and5 that occurred between 1821 and 1910, another from Estonia that occurred in 1670 with a recorded magnitude of 4.7 etc.The smallest event from SNSN in the declustered catalogue is below Mw 0. Classification of events is a major undertaking at the

seismic networks in Fennoscandia, as the rate of detected blasts is 10 times higher than the earthquake rate. Analyst experience during manual review together with a recently developed ML system (Eggertsson et al., 2024) is used to classify events. The process is however out of scope for this paper. We added brief info on the ESHM20 events and classification to 4.1.1.

Line 231: Indicate the magnitude range for the 24,215 events in the final catalogue.
Modified as per reviewer's suggestion.

Section 4.1.2: How do the results of the declustering method (modified Gardner and Knopoff, 1974) compare with that from the method of Burkhard and Grünthal (2009) that was calibrated for the earthquake catalogue in Central Europe and was used in ESHM13 and ESHM20?
As the Uski et al. (2015) method worked well, we have not used the Burkhard & Grunthal (2009) approach.

Line 223: Replace "at our disposal spans the year 1375 until the end of" with "that we used spans between 1375 and the end of".
Stylistic, we keep our text.

Line 234: Provide a reference for the first sentence.
An appropriate reference has now been added

Lines 249-251: It is difficult to follow this sentence, so I would suggest rephrasing it.
The text has been rephrased

Line 257: Replace "a smaller fraction of dependent events of only 11%, a difference to our result which is likely due to the fact that" with "less dependent events than those in our study. The difference (11%) is probably because".
The sentence has been rephrased.

Line 249: How do the earthquakes in the FENCAT compare with those in the ESHM20 catalogue in terms of epicentral location and magnitude? In Figure 4 the earthquakes should be plotted in terms of magnitude to facilitate this comparison. Figure 2: I would suggest adding an extra figure to show the distribution of the seismic source model. Figure 2 should show only the final catalogue for this work where the distribution of earthquakes in terms of magnitude should be highlighted.

ESHM20 used the FENCAT catalogue for most of the northern Europe data. Epicentral locations should therefore be the same (as those in FENCAT when the ESHM20 data was assembled) but homogenized moment magnitudes may vary as the homogenisation schemes are different. We have not done an exhaustive comparison of the ESHM20 event catalogue. We modified Fig 4 such that our events with Mw >= 3.5 are identifiable. It is unclear to us what the reviewer refers to with "an extra figure to show the distribution of the seismic source model". In response to earlier comments we modify Fig 2A such that only shows events with Mw>=4 whereas Fig 2B will continue to show our full, declustered catalogue.

Line 312: Replace "is complicated by" with "is difficult due to".
Stylistic, we keep our text.

Line 315: Delete "purposes".
Modified as per reviewer's suggestion.

Lines 320 and the following lines: Indicate the magnitude scale.
The magnitude scales are now indicated.

Table 2: Why aren't the recurrence parameters of all ASZs reported in this table? The a and b-values for zones 1,4-8,10-12, etc are missing. For transparency, they all should be reported. Is the activity rate computed for 0 Mw? It would be also useful to indicate how many earthquakes within the completeness thresholds were used to estimate the recurrence parameters. As mentioned before, it is unclear which completeness thresholds were used for the estimation of the recurrence parameters. For many zones, the b-value seems to be quite low (< 0.9), what is the reason for this? How do the b-values compare with previous studies, in particular the ESHM20 for similar zones?
We chose to only include recurrence parameters for zones 2, 3, 4, 9, 13, 14, 15, 18, 23, 24, 30 and 31 as these are the only zones that lie within Sweden.  The new table indicates the number of complete events used towards the recurrence parameter calculations. The activity rate is indeed calculated for 0 Mw. Our b-values are generally in line with the ESHM20 b-values, within the uncertainty limits. We have one b-value comparison in section 6.2, we added more comparisons.

Line 383: replace "construct" with "develop".
Stylistic, no change.

Line 415: What is Model 5?
Good catch, that should be Model 4, also on lines 418 and 422. We had five models early on. Changed.

Line 433: Replace "yearly" with "annual".
Changed

Lines 441-444: The hypocentral depths of the earthquakes in the catalogue have not been discussed at all in the manuscript to justify the depth distribution indicated here for the hazard calculations.
As indicated above, we have now improved the text to include a depth distribution discussion.

Section 4.7: Openquake requires also the definition of the faulting style for potential, future earthquakes, defined by rake, dip and strike. This set of parameters has not been discussed at all in the manuscript.
Openquake requires the definition of the strike, dip and rake of the nodal plane orientations. We have defined these nodal planes to be oriented parallel and perpendicular to the regional stress field, as defined in the ESHM20 calculations. The text has been updated to say the

following - "*Each seismic source area requires the definition of orientations and faulting styles of ruptures, quantified by the strike dip and rake of nodal planes. We use the strike and dip values adopted by ESHM20 in their calculations and choose a strike of 0 degrees, a dip of 90 degrees. We define two rake values of 0 or 180 degrees, each with a weight of 0.5 in the probability distribution.*

Line 456: In the revision of the Eurocode 8, the seismic hazard is described in terms of the 5% damped maximum spectral acceleration at a short period and 1.0 s period, and PGA is not mentioned anymore. Would it be useful to estimate national maps also for spectral acceleration for a representative short period (e.g. 0.2 s) and 1.0 s?
We agree and the new hazard maps with Mmin = 4 include results for the spectral acceleration calculations.

Figure 10: What is plotted in Figure 10 exactly? Is this the relative or absolute difference between the new map for Sweden and the ESHM20 maps? It would be helpful to produce such a map also for ESHM20 and the other previous maps discussed in Section 6.2.
Figure 10 shows the difference in mean PGA between our new model and the ESHM20, for the two return periods. Absolute in the sense that it is just our Model – ESHM20, which is why some areas are positive and some negative. As we have focused on a comparison to ESHM20, we choose not to include difference maps of other studies.